# Leveraging Unlabeled Data Sharing through Kernel Function Approximation in Offline Reinforcement Learning

**Yen-Ru Lai**                                           *r09942079@ntu.edu.tw*
*Graduate Institute of Communication Engineering, National Taiwan University*

**Fu-Chieh Chang**                                       *d09942015@ntu.edu.tw*
*Graduate Institute of Communication Engineering, National Taiwan University*
*Mediatek Research*

**Pei-Yuan Wu**                                          *peiyuanwu@ntu.edu.tw*
*Graduate Institute of Communication Engineering, National Taiwan University*

**Reviewed on OpenReview:** `https://openreview.net/forum?id=78N9tCL6Ly`

## Abstract

Offline reinforcement learning (RL) learns policies from a fixed dataset, but often requires large amounts of data. The challenge arises when labeled datasets are expensive, especially when rewards have to be provided by human labelers for large datasets. In contrast, unlabelled data tends to be less expensive. This situation highlights the importance of finding effective ways to use unlabelled data in offline RL, especially when labelled data is limited or expensive to obtain. In this paper, we present the algorithm to utilize the unlabeled data in the offline RL method with kernel function approximation and give the theoretical guarantee. We present various eigenvalue decay conditions of the RKHS $\mathcal{H}_k$ induced by kernel $k$ which determine the complexity of the algorithm. In summary, our work provides a promising approach for exploiting the advantages offered by unlabeled data in offline RL, whilst maintaining theoretical assurances.

## 1 Introduction

Reinforcement learning (RL) algorithms have demonstrated empirical success in a variety of domains, including the defeat of Go champions (Silver et al., 2016), robot control (Kalashnikov et al., 2018), and the development of large language models (Stiennon et al., 2020). In particular, these achievements are largely associated with online reinforcement learning, characterized by dynamic data collection. However, the widespread adoption of online RL faces significant challenges. In many scenarios, active exploration is impractical due to factors such as the high cost of data collection (Levine et al., 2020). To this end, in this paper we explore offline reinforcement learning - a fully data-driven framework similar to supervised learning. Unfortunately, fully data-driven offline RL demands large datasets. In more realistic scenarios, offline reinforcement learning (RL) could allow us to use a smaller amount of task-specific data along with a significant amount of task-agnostic data. This data is not labeled with task rewards, and some of it may not be directly relevant to the task at hand.

Prior works use learned classifiers that discriminate between successes and failures for reward labeling (Fu et al., 2018; Singh et al., 2019) in the online RL setting. However, these approaches are unsuitable for the offline RL setting since they require real-time interaction. Alternatively, some research focuses on learning from data without explicit reward labels by directly imitating expert trajectories (Ho & Ermon, 2016; Kostrikov et al., 2019) or deriving the reward function through inverse reinforcement learning using

an expert dataset (Fu et al., 2017; Finn et al., 2016). However, in real-world scenarios, these approaches may face challenges due to the resource-intensive and costly nature of the expert trajectory acquisition and reward labelling process.

Yu et al. (2022) has revealed the challenges associated with learning to predict rewards, highlighting the surprising efficacy of setting the reward to zero. Despite these findings, the impact of reward prediction methods on performance and the potential demonstrable benefits of reward-free data in offline reinforcement learning (RL) remain unclear. In response to this, Hu et al. (2023) have introduced a novel model-free approach named Provable Data Sharing (PDS). PDS incorporates uncertainty penalties into the learned reward functions, maintaining a conservative algorithm. This method allows PDS to take advantage of unlabeled data for offline RL, especially in linear MDPs. However, the linear MDP assumption is inflexible and rarely is fulfilled in practice. This question naturally arises.

How can we enhance the performance of offline RL algorithms that use kernel function approximation by effectively using reward-free data?

This work focuses on the episodic Markov decision process (MDP). The reward function and value function are both represented by kernel functions. Inspired by the Provable Data Sharing (PDS) (Hu et al., 2023) framework, we propose a new algorithm. The PDS algorithm has two main components. First, it pessimistically estimates rewards by applying additional penalties to the reward function learned from labeled data. This augmentation is designed to prevent overestimation, thus ensuring a conservative algorithm. The second part of the PDS algorithm uses the Pessimistic Value Iteration (PEVI) algorithm introduced by Jin et al. (2021) to derive the policy. Our main contribution is that

- **Extension of PDS framework:** We expand the applicability of the Provable Data Sharing (PDS) framework, initially introduced by Hu et al. (2023). This extension goes beyond the original linear Markov Decision Process (MDP) setting, incorporating kernel function approximation. This expansion enhances the versatility of the PDS framework, making it applicable to a broader range of scenarios. Our derivation is influenced by methodologies proposed for kernelized contextual bandits (Chowdhury & Gopalan, 2017; Valko et al., 2013; Srinivas et al., 2009), as well as techniques such as pessimistic value iteration (PEVI) (Jin et al., 2021) and the kernel optimum least squares value iteration algorithm (KOVI) (Yang et al., 2020).

- **Focus on finite-horizon MDPs:** While Hu et al. (2023) focuses on discounted infinite-horizon MDPs, our work shifts attention to finite-horizon MDPs, which introduce unique challenges due to their horizon-dependent reward and transition functions. Unlike the discounted setting, where strong coverage assumptions such as finite concentrability coefficients are often required, our framework does not rely on such uniform data coverage.

- **Feature coverage assessment via concentratability coefficient:** In contrast to Hu et al. (2023), which relies on a bounded concentratability coefficient to assess coverage over the state-action space, we introduce a global coverage assumption (Assumption 4.7) based on the spectrum of feature covariance matrices. This approach (Wang et al., 2020a), widely used in supervised learning, is particularly well-suited for settings with linear function approximation. Unlike single-policy coverage, our assumption requires sufficient data coverage across all policies in the considered class, ensuring broader applicability.

- **Enhance the suboptimality:** By employing the data-splitting technique discussed by Xie et al. (2021), the suboptimality can be enhanced by a factor of $\sqrt{d}$, which depends on the choice of kernel. This enhancement comes at the cost of $\sqrt{H}$, a constant inherent in the MDP. Nevertheless, with an appropriate selection of kernel, the overall algorithmic performance can be significantly improved.

Our research provides a theoretical guarantee for effectively utilizing the benefits of reward-free data in offline RL. We aim to enhance the robustness of offline RL methods by maintaining theoretical guarantees, which offers a valuable contribution to the ongoing development of more resilient and efficient RL frameworks.

## 2 Related Works

The issue of suboptimality in discounted and episodic MDP with a model has been considered in linear and kernel settings. The results are presented in Table 1. In the episodic MDP setting, we have the dataset with $N$ trajectories of horizon $H$, and the suboptimality dependent on $N$ and $H$. On the other hand, in a discounted MDP setting, we have the dataset with length $N$, and suboptimality dependent on $N$ and the discount factor $\gamma$. The PEVI algorithm (Jin et al., 2021) serves as the foundational algorithm within Hu et al. (2023) and our work. If we assume that the infinite horizon MDP should conclude within $H$ steps (referred to as the effective horizon) (Yan et al., 2022), we can set the discount factor $\gamma$ such that $H = 1/(1 - \gamma)$. Consequently, the suboptimality for the PDS algorithm is expressed as $\tilde{\mathcal{O}}(dH^2 N_2^{-\frac{1}{2}})$ where $N_2$ is the number of trajectories for the unlabeled dataset. Similar to Hu et al. (2023), we incorporate unsupervised data sharing to enhance the offline RL algorithm. The linear setting is a special case of the kernel setting with a linear kernel. In this case, we can recover the suboptimality as $\tilde{\mathcal{O}}(Hd^{\frac{1}{2}} N_1^{-\frac{1}{2}})$, where $N_1$ is the number of trajectories for the labeled dataset, as provided in Hu et al. (2023). A notable difference between PEVI and PDS lies in PDS's utilization of data sharing to improve the suboptimality through an unlabeled dataset. It's important to note that $N_2 > N_1$ in general. When comparing PDS with our approach in a linear setting, the $H$-folds data splitting in our algorithm enhances the suboptimality by a factor of $\sqrt{d}$. However, this improvement comes with a tradeoff, as our algorithm introduces a suboptimality increment by a factor of $\sqrt{H}$ because we need to partition the data set into $H$ folds. As a result, each estimated value function is derived from only $N_2/H$ episodes of data.

Table 1: The existing suboptimality under weak convergence (see Assumption 4.7)(except for the last row), discussed in Section 2. Here, the labeled dataset is represented as $\{(s'^{\tau}_h, a'^{\tau}_h, r^{\tau}_h)\}^{N_1, H}_{\tau, h=1}$, unlabeled dataset is represented as $\{(s'^{\tau+N_1}_h, a'^{\tau+N_1}_h)\}^{N_2, H}_{\tau, h=1}$. Denote $G(N, \lambda)$ as the maximum information gain, $\zeta_{\mathcal{D}} = \max_{h \in [H]} \mathbb{E}_{\pi^*}[\zeta_h(\mathcal{D}', \mathcal{D})| s_1 = s_0]$ represents a maximum amount of information from the dataset $\mathcal{D}$ and $\mathcal{D}'$, where $\mathcal{D}'$ is the combination of $\mathcal{D}$ and observed data $z$, and $\zeta_{\widetilde{\mathcal{D}}} = \max_{h \in [H]} \mathbb{E}_{\pi^*}\left[\zeta_h((\widetilde{\mathcal{D}}^{\theta}_h)', \widetilde{\mathcal{D}}^{\theta}_h)| s_1 = s_0\right]$, where the definition of $\widetilde{\mathcal{D}}^{\theta}_h$ is shown in Theorem 4.3, $\nu = 1 + \frac{1}{N_1}$, $\lambda = 1 + \frac{1}{N}$ and $N = N_1 + N_2$. In a linear MDP setting, it is stated that the transition probability can be represented linearly in a feature map of state-action with $d$ dimensions.

| Algorithm | MDP | Setting | SubOpt |
|---|---|---|---|
| PEVI (Jin et al., 2021) | Episodic | Linear | $\tilde{\mathcal{O}}(dH^2 N_1^{-\frac{1}{2}})$ |
| PDS (Hu et al., 2023) | Discounted | Linear | $\tilde{\mathcal{O}}(d^{\frac{1}{2}}(1-\gamma)^{-1} N_1^{-\frac{1}{2}}) + \tilde{\mathcal{O}}(d(1-\gamma)^{-2} N_2^{-\frac{1}{2}})$ |
| Our work | Episodic | kernel-based, $d$-finite spectrum | $\tilde{\mathcal{O}}(Hd^{\frac{1}{2}} N_1^{-\frac{1}{2}}) + \tilde{\mathcal{O}}(H^{\frac{5}{2}} d^{\frac{1}{2}} N_2^{-\frac{1}{2}})$ |
| Our work | Episodic | kernel-based, general setting | $\tilde{\mathcal{O}}(H\sqrt{G(N_1, \nu)\zeta_{\mathcal{D}_1}}) + \tilde{\mathcal{O}}(H^2 \sqrt{G(\frac{N}{H}, \lambda)\zeta_{\widetilde{\mathcal{D}}}})$ |

### 2.1 Offline Reinforcement Learning

In offline reinforcement learning (RL), the goal is to learn a policy from a static data set collected previously without interacting with the environment. Current approaches in offline RL (Levine et al., 2020) can be broadly classified into dynamic programming methods and model-based methods. Dynamic programming methods aim to learn a state action value function, known as the $Q$ function. Subsequently, this value function is used either to directly find the optimal policy or, in the case of actor-critic methods, to estimate a gradient for the expected returns of a policy. The offline dynamic programming algorithm operates in a tabular setting (Jin et al., 2018). However, algorithms designed for tabular settings have limitations when applied to function approximation settings with a large number of effective states. Recent work has centered around the functional approximation setting, especially in the linear setting, where the value function (or transition model) can be represented using a linear function of a known feature mapping (Jin et al., 2021; Cai et al., 2020; Zanette et al., 2021). As the linear Markov decision process (MDP) assumption is rigid and

rather restrictive in practice, Wang et al. (2020b) explores the kernel optimal least squares value iteration (KOVI) algorithm (Yang et al., 2020) for general function approximation. In contrast, model-based methods rely on their ability to estimate the transition function using a parameterized model, such as a neural network. Instead of employing dynamic programming methods to fit the model, model-based approaches leverage their ability to effectively utilize large and diverse datasets to estimate the transition function (Yu et al., 2021b; Janner et al., 2019; Uehara & Sun, 2021). Both of the methods presented above require a large amount of data to learn a state-action or transition function. In our work, we use reward-free data (i.e., unlabeled data) to improve the performance of learning a state-action function. On the theoretical front, Yin et al. (2022b) explore offline reinforcement learning with differentiable function class approximations, extending to non-linear function approximation. Blanchet et al. (2024) investigate distributionally robust offline reinforcement learning (robust offline RL), which aims to identify an optimal policy from offline datasets that remains effective in perturbed environments. Meanwhile, Hu et al. (2024) addresses the fundamental challenge of transitioning from offline learning to online fine-tuning.

## 2.2 Offline Data Sharing

Data sharing strategies in multi-task reinforcement learning (RL) have shown effectiveness, as observed in works such as Yu et al. (2021a); Eysenbach et al. (2020); Chen et al. (2021). This involves reusing data across different tasks by relabeling rewards, thereby enhancing performance in multi-task offline RL scenarios. Prior work has employed various relabeling strategies. These include uniform labeling (Kalashnikov et al., 2021), labeling based on metrics such as estimated $Q$-values (Yu et al., 2021a), and labeling based on distances to states in goal-conditioned settings (Chen et al., 2021). However, these approaches either necessitate access to the functional form of the reward for relabeling or are confined to goal-conditioned settings. On the other hand, Yu et al. (2022) proposes a straightforward strategy by assigning zero rewards to unlabeled data. On the other hand, Hu et al. (2023) employs linear regression to label rewards for unlabeled data. These approaches present alternative and potentially simpler methods for relabeling, especially in scenarios where direct access to the reward function is challenging or unavailable. In our work, we propose kernel ridge regression to exploit unlabeled data which under certain conditions can be reduced to linear regression.

## 3 Background

### 3.1 Episodic Markov Decision Process

Consider an episodic MDP (Yang et al., 2020; Sutton & Barto, 2018), denoted as $\mathcal{M} = (\mathcal{S}, \mathcal{A}, H, \mathcal{P}, r)$ with state space $\mathcal{S}$, action space $\mathcal{A}$, horizon $H$, transition function $\mathcal{P} = \{\mathcal{P}_h\}_{h \in [H]}$, and reward function $r = \{r_h\}_{h \in [H]}$. We assume that the reward function is bounded, that is, $r_h \in [0, 1]$. For any policy $\pi = \{\pi_h\}_{h \in [H]}$ and $h \in [H]$, we define the state-value function $V_h^\pi : \mathcal{S} \to \mathbb{R}$ and the action-valued function (Q-function) $Q_h^\pi : \mathcal{S} \times \mathcal{A} \to \mathbb{R}$ as $V_h^\pi(s) = \mathbb{E}_\pi \left[ \sum_{t=h}^H r_t(s_t, a_t) | s_h = s \right]$ and $Q_h^\pi(s, a) = \mathbb{E}_\pi \left[ \sum_{t=h}^H r_t(s_t, a_t) | s_h = s, a_h = a \right]$. These two functions satisfy the well-known Bellman equation: $V_h^\pi(s) = \langle Q_h^\pi(s, \cdot), \pi_h(\cdot \mid s) \rangle_{\mathcal{A}}$ and $Q_h^\pi(s, a) = \mathbb{E} \left[ r_h(s_h, a_h) + V_{h+1}^\pi(s_{h+1}) \mid s_h = s, a_h = a \right]$. For any function $f : \mathcal{S} \to \mathbb{R}$, we define the transition operator at each step $h \in [H]$ as $(\mathbb{P}_h f)(s, a) = \mathbb{E} \left[ f(s_{h+1}) \mid s_h = s, a_h = a \right]$, and define the Bellman operator as $(\mathbb{B}_h f)(s, a) = \mathbb{E} \left[ r_h(s_h, a_h) \mid s_h = s, a_h = a \right] + (\mathbb{P}_h f)(s, a)$. Similarly, for all $h \in [H]$, the Bellman optimality equations defined as $V_h^*(s) = \sup_{a \in \mathcal{A}} Q_h^*(s, a)$ and $Q_h^*(s, a) = (\mathbb{B}_h V_{h+1}^*)(s, a)$. Meanwhile, the optimal policy $\pi^*$ satisfies $\pi_h^*(\cdot \mid s) = \operatorname*{argmax}_{\pi_h} \langle Q_h^*(s, \cdot), \pi_h(\cdot \mid s) \rangle_{\mathcal{A}}$ and $V_h^*(s) = \langle Q_h^*(s, \cdot), \pi_h^*(\cdot \mid s) \rangle_{\mathcal{A}}$. Reinforcement learning aims to learn a policy maximizing expected cumulative reward. Accordingly, we define the performance metric(i.e., suboptimality) as

$$\mathrm{SubOpt}(\pi; s) = V_1^{\pi^*}(s) - V_1^\pi(s). \tag{1}$$

### 3.2 Assumption of Offline Data

In offline RL setting, a learner uses pre-collected dataset $\mathcal{D}$, which consists of $N$ trajectories $\left\{ \left( s_h'^\tau, a_h'^\tau, r_h^\tau \right) \right\}_{\tau, h=1}^{N, H}$, generated by some fixed but unknown MDP $\mathcal{M}$ under the behavior policy $\pi^{\mathrm{b}}$ in the following manner: $s_1'^\tau \sim \rho^{\mathrm{b}}, a_h'^\tau \sim \pi_h^{\mathrm{b}} \left( \cdot \mid s_h'^\tau \right)$ and $s_{h+1}'^\tau \sim \mathcal{P}_h \left( \cdot \mid s_h'^\tau, a_h'^\tau \right), 1 \leq h \leq H$. Here $\rho^{\mathrm{b}}$ represents a

predetermined initial state distribution associated with the static dataset. The learner may also have partial observations of the reward in addition to the above state-action observations. More elaborately, we assume access to both a labeled dataset $\mathcal{D}_1 = \left\{\left(s'^\tau_h, a'^\tau_h, r^\tau_h\right)\right\}^{N_1, H}_{\tau, h=1}$, and an unlabeled dataset $\mathcal{D}_2 = \left\{\left(s'^{\tau+N_1}_h, a'^{\tau+N_1}_h\right)\right\}^{N_2, H}_{\tau, h=1}$. We utilize the estimated reward function with parameter $\widetilde{\theta}$, as determined in equation (4), to relabel dataset $\mathcal{D}_2$. The relabeled dataset, denoted as $\mathcal{D}^{\widetilde{\theta}}_2 = \left\{\left(s'^{\tau+N_1}_h, a'^{\tau+N_1}_h, \widetilde{r}^{\widetilde{\theta}_h}_h(s'^{\tau+N_1}_h, a'^{\tau+N_1}_h)\right)\right\}^{N_2, H}_{\tau, h=1}$.

### 3.3 Reproducing Kernel Hilbert Space

Consider a reproducing kernel Hilbert space (RKHS) as a function space. For simplicity, let $z = (s, a)$ denote a state-action pair and denote $\mathcal{Z} = \mathcal{S} \times \mathcal{A}$. Without loss of generality, we regard $\mathcal{Z}$ as a compact subset of $\mathbb{R}^m$, where the dimension $m$ is fixed. Let $k : \mathcal{Z} \times \mathcal{Z} \to \mathbb{R}$ be a positive definite continuous kernel and its corresponding Gram matrix $[K]_{i,j} = k(z_i, z_j), \forall i, j \in [m]$. Note that $K$ is positive semi-definite. Define $\mathcal{H}_k$ as the RKHS induced by $k$, containing a family of functions defined in $\mathcal{Z}$. Let $\langle \cdot, \cdot \rangle_{\mathcal{H}_k} : \mathcal{H}_k \times \mathcal{H}_k \to \mathbb{R}$ and $\| \cdot \|_{\mathcal{H}_k} : \mathcal{H}_k \to \mathbb{R}$ denote the inner product and the norm on $\mathcal{H}_k$, respectively. According to the reproducing property, for all $f \in \mathcal{H}_k$, and $z \in \mathcal{Z}$, holds $f(z) = \langle f, k(\cdot, z) \rangle_{\mathcal{H}_k}$. For more details and different characterizations of RKHS, see Aronszajn (1950); Berlinet & Thomas-Agnan (2011). Without loss of generality, we assume that $\sup_{z \in \mathcal{Z}} k(z, z) \leq 1$.

Let $\mathcal{L}^2(\mathcal{Z})$ be the set of square-integrable functions on $\mathcal{Z}$ with respect to the Lebesgue measure and let $\langle \cdot, \cdot \rangle_{\mathcal{L}^2}$ be the inner product on $\mathcal{L}^2(\mathcal{Z})$. The kernel function $k$ induces an integral operator $T_k : \mathcal{L}^2(\mathcal{Z}) \to \mathcal{L}^2(\mathcal{Z})$ defined as $T_k f(z) = \int_{\mathcal{Z}} k(z, z') f(z') \, dz'$ for all $f \in \mathcal{L}^2(\mathcal{Z})$. By Mercer's theorem (Steinwart & Christmann, 2008), the integral operator $T_k$ has countable and positive eigenvalues $\{\sigma_i\}_{i \geq 1}$ and the corresponding eigenfunctions $\{\psi_i\}_{i \geq 1}$. Then, the kernel function admits a spectral expansion $k(z, z') = \sum^\infty_{i=1} \sigma_i \psi_i(z) \psi_j(z')$. Moreover, the RKHS $\mathcal{H}_k$ can be written as a subset of $\mathcal{L}^2(\mathcal{Z})$ such that $\mathcal{H}_k = \left\{ f \in \mathcal{L}^2(\mathcal{Z}) : \sum^\infty_{i=1}(1/\sigma_i) \langle f, \psi_i \rangle^2_{\mathcal{L}^2} < \infty \right\}$, and the inner product of $\mathcal{H}_k$ also can be written as $\langle f, g \rangle_{\mathcal{H}_k} = \sum^\infty_{i=1}(1/\sigma_i) \langle f, \psi_i \rangle_{\mathcal{L}^2} \langle g, \psi_i \rangle_{\mathcal{L}^2}$ for all $f, g \in \mathcal{H}_k$. With the above construction, the scaled eigenfunctions $\{\sqrt{\sigma_i}\psi_i\}_{i \geq 1}$ form an orthonormal basis for $\mathcal{H}_k$. We define the mapping $\phi : z \mapsto k(z, \cdot)$ to transform data from $\mathcal{Z} = \mathcal{S} \times \mathcal{A}$ to the (possibly infinite-dimensional) RKHS $\mathcal{H}_k$, which satisfies $k(z, z') = \langle \phi(z), \phi(z') \rangle_{\mathcal{H}_k}$ for all $z, z' \in \mathcal{Z}$ (Steinwart & Christmann, 2008, Lemma 4.19). We define the maximum information gain (Srinivas et al., 2009) to describe the complexity of $\mathcal{H}_k$:

$$G(n, \lambda) = \sup \left\{ \frac{1}{2} \log \det (I + K_{\mathcal{D}}/\lambda) : \mathcal{D} \subset \mathcal{Z}, |\mathcal{D}| \leq n \right\}, \tag{2}$$

where $K_{\mathcal{D}}$ is the Gram matrix for the set $\mathcal{D}$. Furthermore, the magnitude of maximal information gain $G(n, \lambda)$ depends on how rapidly the eigenvalues decay to zero, serving as a proxy dimension of $\mathcal{H}$ in the case of an infinite-dimensional space. If $\mathcal{H}_k$ is of finite rank, we have that $G(n, \lambda) = \mathcal{O}(d \log n)$ (Yang et al., 2020), where $d$ is the rank of $\mathcal{H}_k$ – referred as the $d$-finite spectrum. In the following, we present several conditions that are often used in the analysis of the RKHS property of $\mathcal{H}_k$ (Yang et al., 2020; Vakili et al., 2021; Yeh et al., 2023) characterizing the eigenvalue decay of $\mathcal{H}_k$.

**Assumption 3.1.** *The integral operator $T_K$ has eigenvalues $\{\sigma_j\}_{j \geq 1}$ and the associated eigenfunctions $\{\psi_j\}_{j \geq 1}$. We assume that $\{\sigma_j\}_{j \geq 1}$ satisfies one of the following conditions for some constant $d > 0$.*

- *$d$-finite spectrum: $\sigma_j = 0, \forall j > d$, where $d$ is a positive integer.*

- *$d$-exponential decay: there exists some constants $C_1, C_2 > 0$ such that $\sigma_j \leq C_1 \cdot \exp\left(-C_2 \cdot j^d\right)$, $\forall j \geq 1$, where $d > 0$.*

- *$d$-polynomial decay: there exists some constants $C_1 > 0$ such that $\sigma_j \leq C_1 \cdot j^{-d}$ $\forall j \geq 1,$, where $d > 1$.*

*For both $d$-exponential decay and $d$-polynomial decay, we assume that there exists $C_\psi > 0$ such that $\sup_{z \in \mathcal{Z}} \sigma^\tau_j \cdot |\psi_j(z)| \leq C_\psi$ holds for all $j \geq 1$ and $\tau \in [0, 1/2)$.*

For instance, let $\mathcal{Z} \subset \mathbb{R}^d$ and assume the kernel function $k(z, z') \leq 1$, the Squared Exponential kernel, defined as

$$k(z, z') = \exp\left(-\frac{|z - z'|^2}{2l^2}\right) \tag{3}$$

where $l$ is a lengthscale parameter, exhibits $d$-exponential decay. Similarly, the Matérn kernel, given by

$$k(z, z') = \frac{2^{1-\nu}}{\Gamma(\nu)} r^\nu B_\nu(r), \quad \text{where } r = \frac{\sqrt{2\nu}}{l}|z - z'|,$$

is characterized by $d$-polynomial decay. Here, $\nu$ determines the smoothness of sample paths (with smaller $\nu$ producing rougher paths), and $B_\nu(r)$ is the modified Bessel function. Theorem 5 in Srinivas et al. (2009) provides detailed proof for these kernel properties.

We assume that the Bellman operator maps any bounded function onto a bounded RKHS norm ball, which is a common assumption used in the function approximation (Yang et al., 2020; Jin et al., 2018).

**Assumption 3.2.** *Define the function class $\mathcal{Q}^* = \{f \in \mathcal{H}_k : \|f\|_{\mathcal{H}_k} \leq R_Q H\}$ for some fixed constant $R_Q > 0$. Then, for any $h \in [H]$ and any $Q : \mathcal{S} \times \mathcal{A} \to [0, H]$, it holds that $\mathbb{B}_h V \in \mathcal{Q}^*$ for $V(s) = \max_{a \in \mathcal{A}} Q(s, a)$.*

A sufficient condition for Assumption 3.2 to hold is when $\mathcal{S} = [0, 1]^m$ and that $r_h(\cdot, \cdot), \mathcal{P}_h(s' \mid \cdot, \cdot) \in \{f \in \mathcal{H}_k : \|f\|_{\mathcal{H}_k} \leq 1\}$ for all $h \in [H], \forall s' \in \mathcal{S}$. To see this, suppose this condition holds, then for any integrable $V : \mathcal{S} \to [0, H]$ holds,

$$\|r_h + \mathbb{P}_h V\|_{\mathcal{H}_k} \leq \|r_h\|_{\mathcal{H}_k} + \|\mathbb{P}_h V\|_{\mathcal{H}_k} \leq 1 + \left\|\int_{s' \in \mathcal{S}} \mathcal{P}_h(s'|\cdot, \cdot) V(s') \, ds'\right\|_{\mathcal{H}_k}$$

$$\leq 1 + \int_{s' \in \mathcal{S}} \|\mathcal{P}_h(s'|\cdot, \cdot) V(s')\|_{\mathcal{H}_k} \, ds' = 1 + \int_{s' \in \mathcal{S}} \|\mathcal{P}_h(s'|\cdot, \cdot)\|_{\mathcal{H}_k} \|V(s')\|_{\mathcal{H}_k} \, ds'$$

$$\leq 1 + H \int_{s' \in \mathcal{S}} ds' = H + 1.$$

Note that under the assumptions of measurability and boundedness on the kernel $k$, $\|\mathbb{P}_h V\|_{\mathcal{H}_k} \in \mathcal{H}_k$, which is given in Muandet et al. (2017, section 3.1). Thus, Assumption 3.2 holds with $R_Q = 2$. This assumption is mild and is also used in Yang et al. (2020). Similar assumptions are used in linear MDP's, which are much stricter (Jin et al., 2021; Zanette et al., 2020).

### 3.4 Pessimistic Value Iteration and Kernel Setting

We consider the pessimistic value iteration, i.e., PEVI (Jin et al., 2021) algorithm, described in Algorithm 2, as the backbone algorithm. This is a model-free, theoretically guaranteed offline algorithm. The fundamental insight of PEVI lies in the incorporation of a penalty function, which essentially introduces a sense of pessimism, into the value iteration algorithm. The key challenge to extend PEVI to kernel setting is that the dimension (even effective dimension) of the kernel based model (when interpreted as linear model) is divergent. In addition, we apply the data splitting method (Rashidinejad et al., 2021; Xie et al., 2021). As introduced in Rashidinejad et al. (2021), data splitting makes sure that the estimated value $\widehat{V}_{h+1}$ and estimated Bellman operator $\widehat{\mathbb{B}}_h$ are estimated using different subsets of $\mathcal{D}$, this yields conditional independence that is required in bounding concentration terms of the form $\left(\widehat{\mathbb{B}}_h - \mathbb{B}_h\right)\widehat{V}_{h+1}$, and hence the suboptimality can be reduced by a factor of $\sqrt{d}$. However, applied naively, this data splitting induces one undesired $\sqrt{H}$ factor in the optimality as we need to split $\mathcal{D}$ into $H$ folds and thus each $\mathbb{B}_h$ is estimated using only $N/H$ episodes of data. Further details of the PEVI algorithm can be found in Appendix B.

## 4 Unsupervised Data Sharing

Our algorithm comprises two main components. The first part involves employing kernel ridge regression to learn the reward function using the labeled dataset and constructing the confidence set. Next, to mitigate

overestimation in reward prediction, we construct the pessimism reward parameter $\tilde{\theta}$ within the confidence set. Section 4.1 discusses this in more detail. The second part involves using the pessimistic reward estimator $\tilde{\theta}$ to relabel the entire dataset, which is a combination of the labeled dataset and the relabeled dataset. Following this, we employ the PEVI algorithm with kernel approximation and data splitting (refer to Algorithm 3) to determine the optimal policy. The detailed steps of the algorithm are outlined in Algorithm 1.

---

**Algorithm 1** Data Sharing, Kernel Approximation

---

1: **Data:** Labeled dataset $\mathcal{D}_1$, and unlabeled dataset $\mathcal{D}_2$.
2: **Input:** Parameter $\beta_h(\delta), \delta, B, \nu, \lambda$.
3: Learn the reward function $\widehat{\theta}_1, \cdots, \widehat{\theta}_H$ from $\mathcal{D}_1$ with

$$\widehat{\theta}_h = \underset{\theta_h \in \mathcal{H}_k}{\operatorname{argmin}} \sum_{\tau=1}^{N_1} \left[ r_h^\tau - \widehat{r}_h^{\theta_h}(s_h'^\tau, a_h'^\tau) \right]^2 + \nu \|\theta_h\|_{\mathcal{H}_k}^2.$$

4: Construct the pessimistic reward function with parameter $\tilde{\theta} := \{\tilde{\theta}_h\}_{h=1}^H$ satisfy

$$\tilde{r}_h^{\tilde{\theta}_h}(s,a) = \max \left\{ \left\langle \widehat{\theta}_h, \phi(s,a) \right\rangle_{\mathcal{H}_k} - \beta_h(\delta) \left\| (\Lambda_h^{\mathcal{D}_1})^{-\frac{1}{2}} \phi(s,a) \right\|_{\mathcal{H}_k}, 0 \right\}. \tag{4}$$

5: Annotate the reward in $\mathcal{D}^\theta$ with parameter $\theta = \tilde{\theta}$, where $\mathcal{D}^\theta$ is a combination of the labeled dataset $\mathcal{D}_1$ and the unlabeled dataset $\mathcal{D}_2^\theta$.
6: Learn the policy from the dataset $\widetilde{\mathcal{D}}_h^{\tilde{\theta}}$ using Algorithm 3 in Appendix:

$$\{\widehat{\pi}_h\}_{h=1}^H \leftarrow \operatorname{PEVI}\left( \mathcal{D}^{\tilde{\theta}}, B, \lambda \right).$$

7: **Result:** $\widehat{\pi} = \{\widehat{\pi}_h\}_{h=1}^H$.

---

### 4.1 Pessimistic Reward Estimation

We utilize labeled dataset $\mathcal{D}_1$ to train a reward function $\widehat{r}_h^{\theta_h}$, using it to label the unlabeled data. Assume that the observed reward is generated as $r_h^\tau = r_h(s_h'^\tau, a_h'^\tau) + \epsilon_h^\tau$ where $r_h : (s,a) \mapsto \langle \theta_h^*, \phi(s,a) \rangle_{\mathcal{H}_k}$ satisfies $r_h(s,a) \in [0,1]$ for all $(s,a) \in \mathcal{S} \times \mathcal{A}$, and $\epsilon_h^\tau$ are i.i.d. centered 1-SubGaussian noise. Here $\theta_h^* \in \mathcal{H}_k$ is an unknown parameter, and $\phi : \mathcal{S} \times \mathcal{A} \to \mathcal{H}_k$ is a known feature map defined in Section 3.3. Furthermore, we assume that $\|\theta_h^*\|_{\mathcal{H}_k} \leq \mathcal{S}$. We learn the reward function from labeled data through a kernel ridge regression problem. Using the feature representation, we write

$$\widehat{\theta}_h = \underset{\theta_h \in \mathcal{H}_k}{\operatorname{argmin}} \sum_{\tau=1}^N \left[ r_h^\tau - \widehat{r}_h^{\theta_h}(s_h'^\tau, a_h'^\tau) \right]^2 + \nu \|\theta_h\|_{\mathcal{H}_k}^2, \tag{5}$$

where $\widehat{r}_h^{\theta_h}(s,a) = \langle \phi(s,a), \theta_h \rangle_{\mathcal{H}_k}$ with parameter $\theta_h$. However, this method leads to an overestimation of predicted reward values, as highlighted in Yu et al. (2022). A novel algorithm called Provable Data Sharing (PDS) is introduced in Hu et al. (2023) to mitigate this problem. PDS incorporates uncertainty penalties into the learned reward functions and integrates seamlessly with existing offline RL algorithms in a linear MDP setting. We extend the application of this algorithm to the kernel setting.

To address the problem of overestimating predicted rewards, we analyze the uncertainty in the learned reward function. The previous solution defines the center of the ellipsoidal confidence set:

$$\mathcal{C}_h(\delta) = \left\{ \theta \in \mathcal{H}_k : \left\| \theta - \hat{\theta}_h \right\|_{\Lambda_h^{\mathcal{D}_1}} \leq \beta_h(\delta) \right\}, \tag{6}$$

where $\|\theta\|_{\Lambda_h^{\mathcal{D}_1}}^2 = \left\langle \theta, \Lambda_h^{\mathcal{D}_1} \theta \right\rangle_{\mathcal{H}_k}$ and $\Lambda_h^{\mathcal{D}_1} = \sum_{\tau=1}^{N_1} \phi(s_h'^\tau, a_h'^\tau)\phi(s_h'^\tau, a_h'^\tau)^\top + \nu I_{\mathcal{H}_k}$ is a positive definite operator, and $\beta_h(\delta)$ is its radius which follows the following proposition.

**Proposition 4.1.** *We define $\beta_h(\delta)$ with the labeled data set $\mathcal{D}_1$ by $\beta_h(\delta) = \sqrt{\nu}\mathcal{S} + \sqrt{\log \frac{\det\left[\nu I + K_h^{\mathcal{D}_1}\right]}{\delta^2}}$, where $K_h^{\mathcal{D}_1}$ is the Gram matrix constructed from the dataset $\mathcal{D}_1$ as $\left[K_h^{\mathcal{D}_1}\right]_{\tau,\tau'} = k(z'^\tau_h, z'^{\tau'}_h)$, where $z'^\tau_h = (s'^\tau_h, a'^\tau_h)$ for $\tau, \tau' \in [N_1]$ and for each $h \in [H]$ and $\delta \in (0, 1)$. Then, with probability at least $1 - \delta$ we have $\left\|\widehat{\theta}_h - \theta_h^*\right\|_{\Lambda_h^{\mathcal{D}_1}} \leq \beta_h(\delta)$, where $\widehat{\theta}_h$ is the solution of equation (5). Furthermore, consider the information gain $G(N, \nu)$, defined in equation (2) of the matrix $K_h^{\mathcal{D}_1}$ and set $\nu = 1 + 1/N_1$, $\beta_h(\delta)$ can be rewritten as*

$$\beta_h(\delta) = \sqrt{\nu}\mathcal{S} + \sqrt{2G(N_1, \nu) + 1 + \log\frac{1}{\delta^2}}. \tag{7}$$

*Moreover, define $\mathcal{C}_h(\delta) = \left\{\theta \in \mathcal{H}_k : \left\|\theta - \widehat{\theta}_h\right\|_{\Lambda_h^{\mathcal{D}_1}} \leq \beta_h(\delta)\right\}$, we have $\mathbb{P}\left(\theta_h^* \in C_h(\delta)\right) \geq 1 - \delta$.*

*Proof.* Please refer to Appendix B.1 for detailed proof. $\square$

In Proposition 4.1, the uncertainty of the learned reward function depends on the maximum information gain of the Gram matrix $K_h^{\mathcal{D}_1}$. However, finding the optimal parameter within the confidence set is computationally inefficient. To address this challenge, Hu et al. (2023) proposes an approach that preserves the pessimistic property of the offline algorithm. This method uses pessimistic estimation, allowing the algorithm to remain pessimistic while mitigating computational challenges. Formally, we construct the pessimistic reward function $\tilde{r}_h^{\tilde{\theta}_h}(s, a)$ for the parameter $\tilde{\theta}_h$ as

$$\tilde{r}_h^{\tilde{\theta}_h}(s, a) = \max\left\{\left\langle\widehat{\theta}_h, \phi(s, a)\right\rangle_{\mathcal{H}_k} - \beta_h(\delta)\left\|(\Lambda_h^{\mathcal{D}_1})^{-\frac{1}{2}}\phi(s, a)\right\|_{\mathcal{H}_k}, 0\right\}. \tag{8}$$

The equation (8) is a direct consequence of the following lemma derived from Cauchy-Schwarz inequalities.

**Lemma 4.2.** $\left|\left\langle\theta_h - \widehat{\theta}_h, \phi(s, a)\right\rangle_{\mathcal{H}_k}\right| \leq \beta_h(\delta)\left\|(\Lambda_h^{\mathcal{D}_1})^{-\frac{1}{2}}\phi(s, a)\right\|_{\mathcal{H}_k}$ *for any $\theta_h \in \mathcal{C}_h(\delta), h \in [H]$.*

The equation (8) provides a lower bound for the reward function within the confidence set $\mathcal{C}(\delta)$. When the labeled data is scarce, or when there is a significant shift in the distribution between the labeled and unlabeled data, the confidence interval becomes wider and then the equation (8) degenerates to 0, which is reduced to the UDS algorithm (Yu et al., 2022; Hu et al., 2023).

## 4.2 Theoretical Analysis

The suboptimality of the Algorithm 1 is characterized by the following theorem.

**Theorem 4.3.** *Consider the MDP described in Section 3.1. Under Assumption 3.1 and Assumption 3.2, and suppose the labeled dataset $\mathcal{D}_1$ and unlabeled dataset $\mathcal{D}_2^\theta$ are defined in Section 3.2. Define $\mathcal{D}^{\widetilde{\theta}} = \{(s_h^\tau, a_h^\tau, \tilde{r}_h^{\widetilde{\theta}_h}(s_h^\tau, a_h^\tau))\}_{\tau, h=1}^{N, H}$, which is a combination of labeled dataset $\mathcal{D}_1$ and unlabeled dataset $\mathcal{D}_2^{\widetilde{\theta}}$ with $N = N_1 + N_2$. We partition dataset $\mathcal{D}^{\widetilde{\theta}}$ into $H$ disjoint and equally sized sub dataset $\{\widetilde{\mathcal{D}}_h^{\widetilde{\theta}}\}_{h=1}^H$, where $|\widetilde{\mathcal{D}}_h^{\widetilde{\theta}}| = N_H = N/H$. Let $\mathcal{I}_h = \{N_H \cdot (h - 1) + 1, \ldots, N_H \cdot h\} = \{\tau_{h,1}, \cdots, \tau_{h,N_H}\}$ satisfy $\widetilde{\mathcal{D}}_h^{\widetilde{\theta}} = \{(s_h^\tau, a_h^\tau, r_h^\tau)\}_{\tau \in \mathcal{I}_h}$. We set $\lambda = 1 + \frac{1}{N}, \nu = 1 + \frac{1}{N_1}$ in Algorithm 1, where*

$$\beta_h(\delta) = \begin{cases} \sqrt{1 + \frac{1}{N_1}}\mathcal{S} + \sqrt{C_1 \cdot d \cdot \log N_1 + \log(\frac{1}{\delta^2})} & \text{d-finite spectrum,} \\ \sqrt{1 + \frac{1}{N_1}}\mathcal{S} + \sqrt{C_1 \cdot (\log N_1)^{1+\frac{1}{d}} + \log(\frac{1}{\delta^2})} & \text{d-exponential decay,} \\ \sqrt{1 + \frac{1}{N_1}}\mathcal{S} + \sqrt{C_1 \cdot (N_1)^{\frac{m+1}{d+m}} \cdot \log(N_1) + \log(\frac{1}{\delta^2})} & \text{d-polynomial decay.} \end{cases}$$

$$B = \begin{cases} C_2 \cdot H \cdot \sqrt{d \log(N/\delta)} & \text{d-finite spectrum,} \\ C_2 \cdot H \cdot \sqrt{(\log N/\delta)^{1+1/d}} & \text{d-exponential decay,} \\ C_2 \cdot N^{\frac{m+1}{2(d+m)}} H^{1 - \frac{m+1}{2(d+m)}} \cdot \sqrt{\log(N/\delta)} & \text{d-polynomial decay.} \end{cases}$$

*Here, $C_1, C_2 > 0$ are absolute constants that does not depend on $N_1, N$, nor $H$. Then, for fixed initial state $s_0 \in \mathcal{S}$, with probability $1 - 2\delta$, the policy $\hat{\pi}$ generated by Algorithm 1 satisfies*

$$\text{SubOpt}(\hat{\pi}; s_0) \leq 2 \sum_{h=1}^{H} \beta_h(\delta) \mathbb{E}_{\pi^*} \left[ \|\phi(s_h, a_h)\|_{(\Lambda_h^{\mathcal{D}_1})^{-1}} \mid s_1 = s_0 \right]$$

$$+ 2B \sum_{h=1}^{H} \mathbb{E}_{\pi^*} \left[ \|\phi(s_h, a_h)\|_{(\Lambda_h^{\widetilde{\mathcal{D}_h^\theta}})^{-1}} \mid s_1 = s_0 \right].$$

*Proof.* For a detailed proof, see Appendix B.2. □

Two key terms express the suboptimality bound. The first term is the reward bias introduced by uncertainties in estimating rewards. This term reflects the challenges and inaccuracies associated with predicting or estimating rewards in a given environment. The second term represents the offline algorithm and optimal policy $\pi^*$ error.

**Remark 4.4.** *We use the Lemma C.2 to rewrite the term of $\beta_h(\delta)$ and $B$ in the Theorem 4.3 as $\beta_h(\delta) = \tilde{\mathcal{O}}(\sqrt{G(N_1, 1 + \frac{1}{N_1})})$ and $B = \tilde{\mathcal{O}}(H\sqrt{G(N, 1 + \frac{1}{N})})$.*

By this remark, both $\beta_h(\delta)$ and $B$ depend on the kernel function class. It is worth noting that $\|\phi(s_h, a_h)\|_{(\Lambda_h^{\mathcal{D}})^{-1}}$ can be expressed as an information quantity for the dataset $\mathcal{D}$, as outlined in the following proposition.

**Proposition 4.5.** *For all $h \in [H]$, we partition dataset $\mathcal{D}$ into $H$ disjoint and equally sized sub datasets $\{\widetilde{\mathcal{D}}_h\}_{h=1}^{H}$, where $\widetilde{\mathcal{D}}_h = \{(s_h^\tau, a_h^\tau, r_h^\tau)\}_{\tau \in \mathcal{I}_h}$ with $\mathcal{I}_h = \{N_H \cdot (h-1) + 1, \ldots, N_H \cdot h\} = \{\tau_{h,1}, \cdots, \tau_{h,N_H}\}$ and $N_H = N/H$. Denote the operator $\Phi_h^{\widetilde{\mathcal{D}}_h} : \mathcal{H}_k \to \mathbb{R}^{N_H}$, and $\Lambda_h^{\widetilde{\mathcal{D}}_h} : \mathcal{H}_k \to \mathcal{H}_k$ as*

$$\Phi_h^{\widetilde{\mathcal{D}}_h} = \begin{pmatrix} \phi\left(z_h^{\tau_{h,1}}\right)^\top \\ \vdots \\ \phi\left(z_h^{\tau_{h,N_H}}\right)^\top \end{pmatrix} = \begin{pmatrix} k\left(\cdot, z_h^{\tau_{h,1}}\right)^\top \\ \vdots \\ k\left(\cdot, z_h^{\tau_{h,N_H}}\right)^\top \end{pmatrix}, \ \Lambda_h^{\widetilde{\mathcal{D}}_h} = \lambda \cdot I_{\mathcal{H}} + (\Phi_h^{\widetilde{\mathcal{D}}_h})^\top \Phi_h^{\widetilde{\mathcal{D}}_h}.$$

*Define Gram matrix $K_h^{\widetilde{\mathcal{D}}_h} = \Phi_h^{\widetilde{\mathcal{D}}_h}(\Phi_h^{\widetilde{\mathcal{D}}_h})^\top$. Then, for any $z \in \mathcal{Z}$, we have*

$$\phi(z)^\top (\Lambda_h^{\widetilde{\mathcal{D}}_h})^{-1} \phi(z) \leq 2 \cdot \left[ \log \det \left( I + K_h^{\widetilde{\mathcal{D}}_h'}/\lambda \right) - \log \det \left( I + K_h^{\widetilde{\mathcal{D}}_h}/\lambda \right) \right], \tag{9}$$

*where $\widetilde{\mathcal{D}}_h'$ is the combination of dataset $\widetilde{\mathcal{D}}_h$ and $z$ which satisfies $\Lambda_h^{\widetilde{\mathcal{D}}_h'} = \Lambda_h^{\widetilde{\mathcal{D}}_h} + \phi(z)\phi(z)^\top$.*

*Proof.* For a detailed proof, see Appendix B.3. □

**Remark 4.6.** *In Proposition 4.5, $\mathcal{H}_k$ can be infinite dimensional. However, for the sake of clarity, we represent $\Phi_h^{\widetilde{\mathcal{D}}_h}$ as a matrix and $\phi(z_h^\tau)$ as a column vector for all $\tau \in \mathcal{I}_h$.*

Here, we define

$$\zeta_h(\mathcal{D}', \mathcal{D}) = 2 \left[ \log \det \left( I + K_h^{\mathcal{D}'}/\lambda \right) - \log \det \left( I + K_h^{\mathcal{D}}/\lambda \right) \right], \tag{10}$$

as the maximal information amount between the dataset $\mathcal{D}'$ and $\mathcal{D}$. Proposition 4.5 states that if the training data set is well known about $z$, then equation (10) will be close to zero. On the other hand, if the training data set is not well known about $z$, then equation (10) will be large.

We specialize the $d$-finite spectrum case of Theorem 4.3 under a weak data coverage assumption to better understand the convergence of Algorithm 1.

**Assumption 4.7** (Weak Convergence). *Suppose the dataset $\mathcal{D} = \{(s_h^\tau, a_h^\tau, r_h^\tau)\}_{\tau,h=1}^{N,H}$ consists of $N$ trajectories, for all $h \in [H]$, the trajectories are drawn independently and identically from distributions induced by some fixed behavior policy $\bar{\pi}$ such that there exists a constant $c_{min} > 0$ satisfying $\inf_{\|f\|_{\mathcal{H}_k}=1}\langle f, \mathbb{E}_{\bar{\pi}}\left[\phi(z_h)\phi(z_h)^\top\right] f\rangle \geq c_{min}$ for any $h \in [H]$.*

Intuitively, Assumption 4.7 posits that the collected data should be relatively well distributed throughout the state action space. Notably, Assumption 4.7 shares similarities with other explorability assumptions common in reinforcement learning literature, such as those in Yin et al. (2022a); Wagenmaker & Pacchiano (2023).

**Corollary 4.8** (Well-Explored Dataset). *In the $d$-finite spectrum case, assume that the Assumption 4.7 holds under the same conditions as Theorem 4.3. Then for $N_1 \geq \Omega(\log(dH/\delta))$ and $N \geq H \cdot \Omega(\log(dH/\delta))$, with probability at least $1 - \delta$, we have*

$$\mathrm{SubOpt}(\hat{\pi}; s) \leq 2\beta_h(\delta) \cdot H \cdot c'/\sqrt{N_1} + 2B \cdot H \cdot c'/\sqrt{N_H}$$
$$\leq \tilde{\mathcal{O}}(H\sqrt{\frac{d}{N_1}}) + \tilde{\mathcal{O}}(H^{\frac{5}{2}}\sqrt{\frac{d}{N_2}}). \tag{11}$$

In the $d$-finite spectrum case, a significant difference between our present study and previous work (Hu et al., 2023) lies in the incorporation of factors $\sqrt{d}$ and $\sqrt{H}$, introduced by the implementation of the data splitting technique (Xie et al., 2021). This technique plays a crucial role in the linear case, influencing the overall convergence behavior of the learned policy. If we aim to transform the feature mapping from a dimensionality of $d$ to $d'$, where $d' > d$, the data partitioning method can help mitigate the convergence of the error bound. Finally, we combine the result in Thereorem 4.3, Remark 4.4, and Corollary 4.8 to get Table 1.

**Remark 4.9.** *For Assumption 4.7, the scenarios involving $d$-exponential and $d$-polynomial decay are not generally valid. If Assumption 4.7 holds true, by integrating Lemma B.2, Lemma B.3, and equation (89), we can deduce the explicit form of $\mathrm{SubOpt}(\hat{\pi}; s)$ as*

$$\mathrm{SubOpt}(\hat{\pi}; s) = \begin{cases} \tilde{\mathcal{O}}\left(HN_1^{-\frac{1}{2}}\right) + \tilde{\mathcal{O}}\left(H^{\frac{5}{2}}N_2^{-\frac{1}{2}}\right) & d\text{-exponential decay,} \\ \tilde{\mathcal{O}}\left(HN_1^{-\frac{1}{2}+\frac{m+1}{d+m}}\right) + \tilde{\mathcal{O}}\left(H^{\frac{5}{2}-\frac{m+1}{2(d+m)}}N_2^{-\frac{1}{2}+\frac{m+1}{2(d+m)}}\right) & d\text{-polynomial decay.} \end{cases} \tag{12}$$

*Nonetheless, Assumption 4.7 does not hold under scrutiny. To demonstrate this, let's assume that Assumption 4.7 is true. It means that for every $f$ within the set $\{\|f\|_{\mathcal{H}_k} = 1\}$, it satisfies $\mathbb{E}_{\tilde{\pi}}[\langle f, \phi(z_h)\phi(z_h)^\top f\rangle] \geq c_{\min}$. Then, we express $\phi$ and $f$ as $\phi = \sum_{i=1}^\infty a_i\psi_i$ and $f = \sum_{i=1}^\infty b_i\psi_i$ respectively, where $\{\psi_i\}_{i=1}^\infty$ is orthonormal basis of $\mathcal{H}_k$. Given that $f$ can represent any function satisfying $\|f\|_{\mathcal{H}_k} = 1$, let $f$ be any vector such that $f = b_j\psi_j$ for an arbitrary $j$. Consequently, for all $j$, the expectation $\mathbb{E}_{\tilde{\pi}}[\langle f, \phi(z_h)\phi(z_h)^\top f\rangle] = a_j^2 \geq c_{\min}$ is satisfied, which results in a paradox because the norm should be finite; however, $\|\phi\|_{\mathcal{H}_k} = \sum_{j=1}^\infty a_j^2 \geq \sum_{j=1}^\infty c_{min} = \infty$.*

## 5  Experiments

In the following experiments, we execute Algorithm 1 to obtain the policy $\hat{\pi}$ and the corresponding value function $V_1^\pi(s)$, where $s$ is sampled from the initial distribution $\rho(s)$. For our experiment, we select the Squared Exponential kernel from equation (3) as the kernel function $k(z, z')$. By applying the kernel trick, the pessimistic reward function $\tilde{r}_h^{\tilde{\theta}_h}(s, a)$, as defined in equation (8), can be computed using the following expression:

$$\tilde{r}_h^{\tilde{\theta}_h}(s, a) = k_h^{\mathcal{D}_1}(s, a)^\top \left(K_h^{\mathcal{D}_1} + \nu I\right)^{-1} y_h$$
$$- \beta_h(\delta) \cdot \nu^{-1/2} \cdot \left(k\left((s, a), (s, a)\right) - k_h^{\mathcal{D}_1}(s, a)^\top \left(K_h^{\mathcal{D}_1} + \nu I\right)^{-1} k_h^{\mathcal{D}_1}(s, a)\right)^{1/2},$$

where $k_h^{\widetilde{\mathcal{D}}_h}$ and $y_h$ defined in equation (17) and (18), respectively. We implemented our algorithm using Python's NumPy library. To ensure reproducibility, our experimental code is publicly accessible[1].

---

[1]https://github.com/d09942015ntu/leveraging_unlabeled_offline_rl

## 5.1 Asymptotic Behavior of $V_1^\pi(s)$

In this experiment, we investigate the asymptotic behavior of $V_1^\pi(s)$. Although we are unable to theoretically validate the correctness of equation (12) due to the violation of Assumption 4.7, we provide empirical evidence that, when the kernel function is the Squared Exponential kernel satisfying $d$-exponential decay, its asymptotic behavior closely aligns with equation (12). We create an toy example of RL environment which meets Assumption 3.2, ensuring bounded RKHS norms for $r_h(\cdot, \cdot)$ and $\mathcal{P}(s'|\cdot, \cdot)$. The MDP $\mathcal{M} = (\mathcal{S}, \mathcal{A}, H, \mathcal{P}, r)$ for this toy example is as follows.

$$\mathcal{S} = [0, C], \quad \mathcal{A} = \{0, 1, ..., C\}, \quad H = C,$$

$$\mathcal{P}_h(s' \mid s, a) = \exp\big[-\alpha \ (s' - ((s + a) \bmod C))^2\big]\Big/\sqrt{\pi/\alpha},$$

$$r(s, a) = \exp\big[-\alpha \ (s - C/2)^2\big]\Big/\sqrt{\pi/\alpha},$$

where $\alpha$ and $C$ are constants, set to $\alpha = 3$ and $C = 8$ in our experiment. To ensure that $s'$ remains within the state space $\mathcal{S}$, we replace $s'$ by "$s' \bmod C$" after sampling $s' \sim \mathcal{P}_h(\cdot|s, a)$. To examine the asymptotic behavior of $V_1^\pi(s)$, we varied the parameters $N_1$ and $N_2$ and plotted the resulting values of $V_1^\pi(s)$ in the left column of Fig. 1. For asymptotic approximation, using equation (12), $V_1^\pi(s)$ can be expressed in the following form:

$$V_1^\pi(s) = c_0 - c_1 N_1^{-\frac{1}{2}} - c_2 N_2^{-\frac{1}{2}},$$

where $c_0, c_1$, and $c_2$ are constants. By applying linear regression to the experimental data (left column of Fig. 1), we estimated these constants and plotted the corresponding curve in the right column of Fig. 1. The strong agreement between the experimental results and the asymptotic approximation validates our hypothesis that the asymptotic behavior of $V_1^\pi(s)$ conforms to our theoretical predictions.

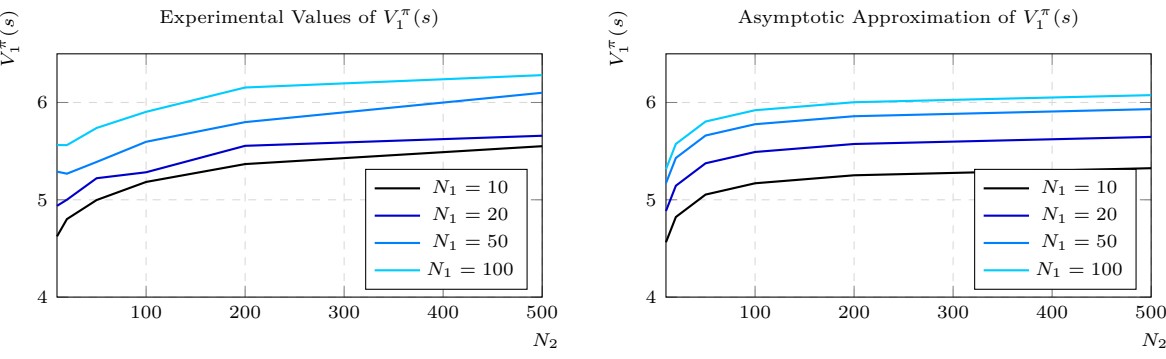

Figure 1: Comparison of experimental values and asymptotic approximation of $V_1^\pi(s)$.

## 5.2 Comparison between Finite Dimensional Features and Kernel Features

In this experiment, we present a real-world example demonstrating the superior performance of our kernel-based offline RL method compared to finite-dimensional feature representations $\phi$, which follows the setting of Hu et al. (2023) but in a finite-horizon scenario. We select the CartPole environment from OpenAI Gym[2] as our test environment. To implement the $\phi$ with finite-dimensional feature representations, we consider three different realizations of $\phi(s, a)$—linear, quadratic, and cubic—denoted as $\phi_{\text{lin}}$, $\phi_{\text{quad}}$, and $\phi_{\text{cub}}$, respectively, which are defined as follows:

$$\phi_{\text{lin}} = (1, x_1, x_2, \cdots, x_n) \quad \text{where } x_1, \cdots, x_n \text{ is the elements in the concatenation of } (s, a)$$

$$\phi_{\text{quad}} = (1, x_1, x_2, \cdots, x_n, x_1^2, x_1 x_2, \cdots, x_{n-1} x_n, x_n^2)$$

$$\phi_{\text{cub}} = (1, x_1, x_2, \cdots, x_n, x_1^2, x_1 x_2, \cdots, x_{n-1} x_n, x_n^2, x_1^3, x_1^2 x_2, x_1 x_2 x_3, \cdots, x_{n-2} x_{n-1} x_n^2, x_{n-1} x_n^2, x_n^3)$$

---

[2]https://www.gymlibrary.dev/environments/classic_control/cart_pole/

We compare these three finite-dimensional feature representations with our kernel-based RL method using the Squared Exponential kernel feature, denoted as $\phi_K$, under various configurations of $N_1$ and $N_2$. The results are presented in Fig. 2. Our findings suggest that, the kernel-based $\phi_K$ generally outperforms alternative finite-dimensional $\phi$ approaches when $N_2 \leq 500$. The performance gap between $\phi_K$ and other $\phi$ becomes more pronounced when $N_2 \leq 100$. This indicates that kernel methods, with their more flexible function approximation, better capture the reward and transition dynamics—particularly when unlabeled data is scarce—resulting in higher $V_1^\pi(s)$ compared to finite-dimensional $\phi$ representations.

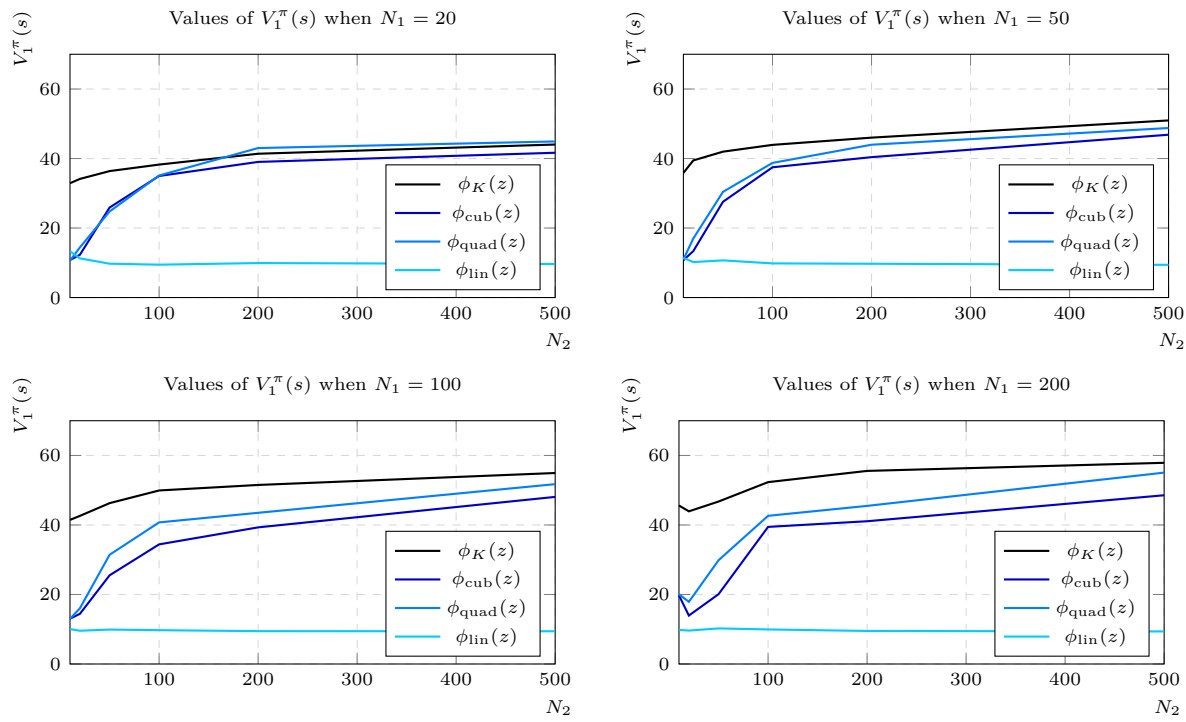

Figure 2: Comparison the values of $V_1^\pi(s)$ between finite dimensional features and kernel features.

# 6 Conclusion

In this paper, we demonstrate that incorporating unlabeled data into offline RL can greatly improve offline RL performance. Our theoretical analysis shows how unlabeled data can improve the performance of offline RL, especially in a more general function approximation setting, in contrast to the results in Hu et al. (2023). Our analysis is based on the common offline RL assumption about the dataset, providing a comprehensive examination of the algorithm's performance under these conditions. In future work, it may be interesting to extend to the discounted MDP setting to deal with more category problems and the low-rank MDP (Uehara et al., 2021).

# Acknowledgments

This work was supported in part by the Ministry of Education (MOE) of Taiwan under Grant NTU-111L891406, the Asian Office of Aerospace Research and Development (AOARD) under Grant NTU-112HT911020, the Center of Data Intelligence: Technologies, Applications, and Systems, National Taiwan University (grant nos. 111L900901/111L900902/111L900903), from the Featured Areas Research Center Program within the framework of the Higher Education Sprout, the Ministry of Education (MOE) of Taiwan, and the financial supports from the Featured Area Research Center Program within the framework of the Higher Education Sprout Project by the Ministry of Education (111L900901/111L900902/111L900903).

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

# A    Pessimistic Value Iteration

The Pessimistic Value Iteration (Jin et al., 2021) (PEVI) algorithm constructs an estimated Bellman operator $\widehat{\mathbb{B}}_h$ based on the dataset $\mathcal{D}$ so that $\widehat{\mathbb{B}}_h \widehat{V}_{h+1}^{\mathcal{D}} : \mathcal{S} \times \mathcal{A} \to \mathbb{R}$ approximates $\mathbb{B}_h \widehat{V}_{h+1}^{\mathcal{D}} : \mathcal{S} \times \mathcal{A} \to \mathbb{R}$. Here $\widehat{V}_{h+1}^{\mathcal{D}} : \mathcal{S} \to \mathbb{R}$ is an estimated value function based on $\mathcal{D}$. Define an uncertainty quantifier with the confidence parameter $\xi \in (0, 1)$ as follows.

**Definition A.1** ($\xi$-Uncertainty Quantifier). *We say $\{\Gamma_h\}_{h=1}^H$ ($\Gamma_h : \mathcal{S} \times \mathcal{A} \to \mathbb{R}$) is a $\xi$-uncertainty quantifier if the event*

$$\mathcal{E} = \left\{ \left| \left( \widehat{\mathbb{B}}_h \widehat{V}_{h+1}^{\mathcal{D}} \right)(s, a) - \left( \mathbb{B}_h \widehat{V}_{h+1}^{\mathcal{D}} \right)(s, a) \right| \leq \Gamma_h(s, a), \quad \forall (s, a) \in \mathcal{S} \times \mathcal{A}, h \in [H] \right\}, \tag{13}$$

*satisfies* $\mathbb{P}(\mathcal{E}) \geq 1 - \xi$.

---

**Algorithm 2** Pessimistic Value Iteration (PEVI): General MDP

1: **Input:** Dataset $\mathcal{D} = \{(s_h^\tau, a_h^\tau, r_h^\tau)\}_{\tau, h=1}^{K, H}$.

2: Initialization: Set $\widehat{V}_{H+1}^{\mathcal{D}}(\cdot) \leftarrow 0$.

3: **for** step $h = H, H - 1, \cdots, 1$ **do**

4:     Construct $\left( \widehat{\mathbb{B}}_h \widehat{V}_{h+1}^{\mathcal{D}} \right)(\cdot, \cdot)$ and $\Gamma_h(\cdot, \cdot)$ based on $\mathcal{D}$.

5:     Set $\bar{Q}_h^{\mathcal{D}}(\cdot, \cdot) \leftarrow \left( \widehat{\mathbb{B}}_h \widehat{V}_{h+1} \right)(\cdot, \cdot) - \Gamma_h(\cdot, \cdot)$.

6:     Set $\widehat{Q}_h^{\mathcal{D}}(\cdot, \cdot) \leftarrow \min \left\{ \bar{Q}_h^{\mathcal{D}}(\cdot, \cdot), H - h + 1 \right\}^+$.

7:     Set $\widehat{\pi}_h(\cdot \mid s) \leftarrow \arg\max_{\pi_h} \left\langle \widehat{Q}_h^{\mathcal{D}}(s, \cdot), \pi_h(\cdot \mid s) \right\rangle_{\mathcal{A}}$.

8:     Set $\widehat{V}_h^{\mathcal{D}}(\cdot) \leftarrow \left\langle \widehat{Q}_h^{\mathcal{D}}(\cdot, \cdot), \widehat{\pi}_h(\cdot \mid \cdot) \right\rangle_{\mathcal{A}}$.

9: **end for**

10: **Output:** $\text{Pess}(\mathcal{D}) = \{\widehat{\pi}_h\}_{h=1}^H, \{\widehat{V}_h^{\mathcal{D}}\}_{h=1}^H$.

---

By equation (13), $\Gamma_h$ quantifies the uncertainty, which allows us to develop the meta-algorithm in Algorithm 2. Now we introduce the PEVI with the data splitting and the kernel setting. Suppose we have the dataset

---

**Algorithm 3** PEVI: Kernel Approximation with Data Split

1: **Input:** Dataset $\mathcal{D} = \{(s_h^\tau, a_h^\tau, r_h^\tau)\}_{\tau, h=1}^{K, H}$ and parameter $B$ and $\lambda$.

2: Data split: Randomly split dataset $\mathcal{D}$ into $H$ disjoint and equally sub-datasets $\{\widetilde{\mathcal{D}}_h\}_{h=1}^H$.

3: Initialization: Set $\widehat{V}_{H+1}^{\widetilde{\mathcal{D}}_{H+1:H}}(\cdot) \leftarrow 0$.

4: **for** step $h = H, \cdots, 1$ **do**

5:     Compute the Gram matrix $K_h^{\widetilde{\mathcal{D}}_h}$, function $k_h^{\widetilde{\mathcal{D}}_h}$ and response vector $y_h$ defined in equation (17) and (18), respectively.

6:     Set $\Gamma_h(\cdot, \cdot) \leftarrow B \cdot \lambda^{-1/2} \cdot \left( k(\cdot, \cdot; \cdot, \cdot) - k_h^{\widetilde{\mathcal{D}}_h}(\cdot, \cdot)^\top \left( K_h^{\widetilde{\mathcal{D}}_h} + \lambda I \right)^{-1} k_h^{\widetilde{\mathcal{D}}_h}(\cdot, \cdot) \right)^{1/2}$.

7:     Set $\bar{Q}_h^{\widetilde{\mathcal{D}}_{h:H}}(\cdot, \cdot) \leftarrow k_h^{\widetilde{\mathcal{D}}_h}(\cdot, \cdot)^\top \left( K_h^{\widetilde{\mathcal{D}}_h} + \lambda I \right)^{-1} y_h - \Gamma_h(\cdot, \cdot)$.

8:     Set $\widehat{Q}_h^{\widetilde{\mathcal{D}}_{h:H}}(\cdot, \cdot) \leftarrow \min \left\{ \bar{Q}_h^{\widetilde{\mathcal{D}}_{h:H}}(\cdot, \cdot), H - h + 1 \right\}^+$.

9:     Set $\widehat{\pi}_h(\cdot \mid s) \leftarrow \arg\max_{\pi_h} \left\langle \widehat{Q}_h^{\widetilde{\mathcal{D}}_{h:H}}(s, \cdot), \pi_h(\cdot \mid s) \right\rangle_{\mathcal{A}}$.

10:    Set $\widehat{V}_h^{\widetilde{\mathcal{D}}_{h:H}}(\cdot) \leftarrow \left\langle \widehat{Q}_h^{\mathcal{D}}(\cdot, \cdot), \widehat{\pi}_h(\cdot \mid \cdot) \right\rangle_{\mathcal{A}}$.

11: **end for**

12: **Output:** $\text{Pess}(\mathcal{D}) = \{\widehat{\pi}_h\}_{h=1}^H, \{\widehat{V}_h^{\widetilde{\mathcal{D}}_{h:H}}\}_{h=1}^H$.

---

$\mathcal{D} = \{(s_h^\tau, a_h^\tau, r_h^\tau)\}_{h, \tau=1}^{H, N}$, we partition the dataset $\mathcal{D}$ into $H$ disjoint and equally sub-datasets $\{\widetilde{\mathcal{D}}_h\}_{h=1}^H$ and

$|\widetilde{\mathcal{D}}_h| := N_H = N/H$ for all $h \in [H]$. Consider the index set $\mathcal{I}_h = \{N_H \cdot (h-1) + 1, \cdots, N_H \cdot h\} = \{\tau_{h,1}, \cdots, \tau_{h,N_H}\}$ such that $\widetilde{\mathcal{D}}_h = \{(s_h^\tau, a_h^\tau, r_h^\tau)\}_{\tau \in \mathcal{I}_h}$. To simplify notation, let $\widetilde{\mathcal{D}}_{h:H} = \bigcup_{t=h}^H \widetilde{\mathcal{D}}_t$ for $h \in [H]$. Note that $\mathcal{D}_{h:H} = \emptyset$ if $h > H$. Then, we construct the pessimistic value iterations (Jin et al., 2021) with $\widehat{\mathbb{B}}_h \widehat{V}_{h+1}^{\widetilde{\mathcal{D}}_{h+1:H}}, \Gamma_h$, and $\widehat{V}_h^{\widetilde{\mathcal{D}}_{h:H}}$. Note that $\widehat{V}_{H+1}^{\widetilde{\mathcal{D}}_{H+1:H}} = 0$. Define the empirical mean squared Bellman error (MSBE) as

$$M_h(f) = \sum_{\tau \in \mathcal{I}_h} \left( r_h^\tau + \widehat{V}_{h+1}^{\widetilde{\mathcal{D}}_{h+1:H}} \left( s_{h+1}^\tau \right) - f \left( s_h^\tau, a_h^\tau \right) \right)^2, \tag{14}$$

at each step $h \in [H]$ and for all $f \in \mathcal{H}_k$. Corresponding, we set

$$\left( \widehat{\mathbb{B}}_h \widehat{V}_{h+1}^{\widetilde{\mathcal{D}}_{h+1:H}} \right)(z) = \widehat{f}_h(z), \quad \text{where } \widehat{f}_h = \underset{f \in \mathcal{H}_k}{\arg\min} \, M_h(f) + \lambda \cdot \|f\|_{\mathcal{H}_k}^2, \tag{15}$$

for $\lambda > 0$. Moreover, we construct $\Gamma_h$ via

$$\Gamma_h(z) = B \cdot \lambda^{-1/2} \cdot \left( k(z,z) - k_h^{\widetilde{\mathcal{D}}_h}(z)^\top \left( K_h^{\widetilde{\mathcal{D}}_h} + \lambda I \right)^{-1} k_h^{\widetilde{\mathcal{D}}_h}(z) \right)^{1/2}, \tag{16}$$

where $B > 0$ is a scaling parameter. Note that it is a bonus function defined in (Yang et al., 2020) and that it is clearly a $\xi$ quantifier. Here, the Gram matrix $K_h^{\widetilde{\mathcal{D}}_h} \in \mathbb{R}^{N_H \times N_H}$, and the function $k_h^{\widetilde{\mathcal{D}}_h} : \mathcal{Z} \to \mathbb{R}^{N_H}$ as

$$\left[ K_h^{\widetilde{\mathcal{D}}_h} \right]_{\tau,\tau'} = k \left( z_h^\tau, z_h^{\tau'} \right), \quad k_h^{\widetilde{\mathcal{D}}_h}(z) = \begin{pmatrix} k \left( z_h^{\tau_{h,1}}, z \right) \\ \vdots \\ k \left( z_h^{\tau_{h,N_H}}, z \right) \end{pmatrix} \in \mathbb{R}^{N_H}, \tag{17}$$

for $\tau, \tau' \in \mathcal{I}_h$. The entry of $y_h \in \mathbb{R}^{N_H}$ corresponding to $\tau \in \mathcal{I}_h$ is

$$[y_h]_\tau = r_h^\tau + \widehat{V}_{h+1}^{\widetilde{\mathcal{D}}_{h+1:H}} \left( s_{h+1}^\tau \right). \tag{18}$$

We construct the pessimistic value iteration with kernel approximation with kernel $k$ by

$$\bar{Q}_h^{\widetilde{\mathcal{D}}_{h+1:H}}(z) = \widehat{\mathbb{B}}_h \widehat{V}_{h+1}^{\widetilde{\mathcal{D}}_{h+1:H}}(z) - \Gamma_h(z),$$

$$\widehat{Q}_h^{\widetilde{\mathcal{D}}_{h+1:H}}(z) = \min \left\{ \bar{Q}_h^{\widetilde{\mathcal{D}}_{h+1:H}}(z), H - h + 1 \right\}^+,$$

$$\widehat{\pi}_h(\cdot \mid s) = \underset{\pi_h}{\arg\max} \left\langle \widehat{Q}_h^{\widetilde{\mathcal{D}}_{h+1:H}}(s, \cdot), \widehat{\pi}_h(\cdot \mid s) \right\rangle_{\mathcal{A}},$$

$$\widehat{V}_h^{\widetilde{\mathcal{D}}_{h+1:H}}(s) = \left\langle \widehat{Q}_h^{\widetilde{\mathcal{D}}_{h+1:H}}(s, \cdot), \widehat{\pi}_h(\cdot \mid s) \right\rangle_{\mathcal{A}}.$$

The algorithm 3 summarizes the entire PEVI algorithm with data splitting.

# B Proof of Main Result

## B.1 Proof of Proposition 4.1

We present a generalization of (Abbasi-Yadkori et al., 2011) (Theorem 1). Its proof closely mirrors that of the special case where $\mathcal{H}_k$ has a linear kernel. To simplify notation, denote the labeled dataset as $\mathcal{D}_1 = \{s_h^\tau, a_h^\tau, r_h^\tau\}_{\tau,h=1}^{N_1,H}$. Subsequently, we address the following problem:

$$\widehat{\theta}_h^t \in \underset{\theta \in \mathcal{H}_k}{\arg\min} \left\{ \sum_{\tau=1}^t \left( \langle \theta, \phi(s_h^\tau, a_h^\tau) \rangle_{\mathcal{H}_k} - r_h^\tau \right)^2 + \nu \|\theta\|_{\mathcal{H}}^2 \right\}. \tag{19}$$

Here, $\phi(s_h^\tau, a_h^\tau)$ is a column vector for all $h \in [H]$ and $t > 0$. The solution for the above equation is that $\widehat{\theta}_h^t = (\Lambda_h^t)^{-1}(\Phi_h^t)^\top Y_h^t$, where $\Lambda_h^t = (\Phi_h^t)^\top \Phi_h^t + \nu I_\mathcal{H}$, $\Phi_h^t = \left[\phi\left(s_h^1, a_h^1\right)^\top, \cdots, \phi\left(s_h^t, a_h^t\right)^\top\right]^\top$, and $Y_h^t = \left[r_h^1, \cdots, r_h^t\right]$. Denote $E_h^t = \left[\epsilon_h^1, \cdots, \epsilon_h^t\right]$ and $K_h^t = \Phi_h^t(\Phi_h^t)^\top$. We then determine the upper bound with $\|\widehat{\theta}_h^t - \widehat{\theta}_h^*\|_{\Lambda_h^t}$. We write

$$
\begin{aligned}
\widehat{\theta}_h^t &= ((\Phi_h^t)^\top \Phi_h^t + \nu I_\mathcal{H})^{-1}(\Phi_h^t)^\top(\Phi_h^t \theta_h^* + E_h^t) \\
&= ((\Phi_h^t)^\top \Phi_h^t + \nu I_\mathcal{H})^{-1}(((\Phi_h^t)^\top \Phi_h^t + \nu I_\mathcal{H}) - \nu I_\mathcal{H})\theta_h^* + (\Phi_h^t)^\top E_h^t) \\
&= ((\Phi_h^t)^\top \Phi_h^t + \nu I_\mathcal{H})^{-1}(\Phi_h^t)^\top E_h^t + \theta^* - \nu((\Phi_h^t)^\top \Phi_h^t + \nu I_\mathcal{H})^{-1}\theta^* \\
&= (\Lambda_h^t)^{-1}(\Phi_h^t)^\top E_h^t + \theta^* - \nu(\Lambda_h^t)^{-1}\theta^*.
\end{aligned}
\tag{20}
$$

That implies

$$
\begin{aligned}
x^\top \hat{\theta}_t - x^\top \hat{\theta}^* &= x^\top (\Lambda_h^t)^{-1}(\Phi_h^t)^\top E_h^t - \nu x^\top (\Lambda_h^t)^{-1}\theta^* \\
&= \langle x, (\Phi_h^t)^\top E_h^t \rangle_{(\Lambda_h^t)^{-1}} - \nu \langle x, \theta^* \rangle_{(\Lambda_h^t)^{-1}},
\end{aligned}
\tag{21}
$$

where $\langle x, y \rangle_{(\Lambda_h^t)^{-1}} = x^\top (\Lambda_h^t)^{-1} y$ and $\Lambda_h^t$ is positive definite, then $(\Lambda_h^t)^{-1/2} \le \nu^{-1/2}$. Using Cauchy-Schwarz inequality, we get

$$
\begin{aligned}
|x^\top \widehat{\theta}_h^t - x^\top \theta_h^*| &\le \|x\|_{(\Lambda_h^t)^{-1}}(\|(\Phi_h^t)^\top E_h^t\|_{(\Lambda_h^t)^{-1}} + \nu \|\theta_h^*\|_{(\Lambda_h^t)^{-1}}) \\
&\le \|x\|_{(\Lambda_h^t)^{-1}}(\|(\Phi_h^t)^\top E_h^t\|_{(\Lambda_h^t)^{-1}} + \sqrt{\nu}\|\theta_h^*\|_{\mathcal{H}_k}),
\end{aligned}
\tag{22}
$$

where the second inequality uses the fact that

$$
\|\theta_h^*\|_{(\Lambda_h^t)^{-1}} = \left\|(\Lambda_h^t)^{-1/2}\theta_h^*\right\|_{\mathcal{H}_k} \le \nu \frac{1}{\sqrt{\nu}}\|\theta_h^*\|_{\mathcal{H}_k} \le \sqrt{\nu}\|\theta_h^*\|_{\mathcal{H}_k}.
$$

Next, we show that with probability at least $1 - \delta$

$$
\|(\Phi_h^t)^\top E_h^t\|_{(\Lambda_h^t)^{-1}}^2 \le H^2 \cdot \log \det\left[\nu I + K_h^t\right] + 2H^2 \cdot \log(1/\delta).
\tag{23}
$$

Following (Valko et al., 2013), we will use the following identities:

$$
\begin{aligned}
\left((\Phi_h^t)^\top \Phi_h^t + \nu I_\mathcal{H}\right)(\Phi_h^t)^\top &= (\Phi_h^t)^\top \left(\Phi_h^t(\Phi_h^t)^\top + \nu I\right) \\
\Rightarrow \qquad \Lambda_h^t (\Phi_h^t)^\top &= (\Phi_h^t)^\top \left(K_h^t + \nu I\right) \\
\Rightarrow \qquad (\Phi_h^t)^\top \left(K_h^t + \nu I\right)^{-1} &= (\Lambda_h^t)^{-1}(\Phi_h^t)^\top.
\end{aligned}
$$

With the basic operation, we get

$$
\begin{aligned}
\|(\Phi_h^t)^\top E_h^t\|_{(\Lambda_h^t)^{-1}}^2 &= (E_h^t)^\top \Phi_h^t (\Lambda_h^t)^{-1}(\Phi_h^t)^\top E_h^t \\
&= (E_h^t)^\top \Phi_h^t(\Phi_h^t)^\top \left(K_h^t + \nu I\right)^{-1} E_h^t \\
&= (E_h^t)^\top K_h^t \left(K_h^t + \nu I\right)^{-1} E_h^t.
\end{aligned}
\tag{24}
$$

Setting $\nu = 1 + \eta$, for some $\eta > 0$, we have

$$
\begin{aligned}
\left(K_h^t + \eta \cdot I\right)\left[K_h^t + (1+\eta)\cdot I\right]^{-1} &= \left(K_h^t + \eta \cdot I\right)\left[I + \left(K_h^t + \eta \cdot I\right)\right]^{-1} \\
&= \left[\left(K_h^t + \eta \cdot I\right)^{-1} + I\right]^{-1},
\end{aligned}
\tag{25}
$$

which implies

$$
\begin{aligned}
(E_h^t)^\top K_h^t \left(K_h^t + \nu I\right)^{-1} E_h^t &\le (E_h^t)^\top \left(K_h^t + \eta \cdot I\right)\left(K_h^t + (1+\eta)I\right)^{-1} E_h^t \\
&= (E_h^t)^\top \left[\left(K_h^t + \eta \cdot I\right)^{-1} + I\right]^{-1} E_h^t.
\end{aligned}
\tag{26}
$$

Applying Lemma C.1 with $\sigma^2 = 1$, we obtain the following result: with a probability of at least $1 - \delta$, the given condition holds simultaneously for all $t \geq 1$

$$(E_h^t)^\top \left[ (K_h^t + \eta \cdot I)^{-1} + I \right]^{-1} E_h^t \leq \log \det \left[ (1+\eta) \cdot I + K_h^t \right] + 2\log(1/\delta),$$

for any $\eta > 0$ and $\delta \in (0,1)$. Combine equation (24) and (26), for any $t > 0$, we get

$$\|(\Phi_h^t)^\top E_h^t\|_{(\Lambda_h^t)^{-1}} \leq \sqrt{\log \frac{\det \left[ (1+\eta) \cdot I + K_h^t \right]}{\delta^2}}, \tag{27}$$

with probability at least $1 - \delta$. Therefore, combine the equation (22) and (27), one also has

$$|x^\top \widehat{\theta}_h^t - x^\top \theta_h^*| \leq \|x\|_{(\Lambda_h^t)^{-1}} \Big( \sqrt{\log \frac{\det \left[ (1+\eta) \cdot I + K_h^t \right]}{\delta^2}} + \sqrt{\nu}\|\theta_h^*\|_{\mathcal{H}_k} \Big), \tag{28}$$

for all $t \geq 0$. Plugging in $x = \Lambda_h^t(\widehat{\theta}_h^t - \theta_h^*)$, we get

$$\|\widehat{\theta}_h^t - \theta_h^*\|_{\Lambda_h^t}^2 \leq \|\Lambda_h^t(\widehat{\theta}_h^t - \theta_h^*)\|_{(\Lambda_h^t)^{-1}} \Big( \sqrt{\log \frac{\det \left[ (1+\eta) \cdot I + K_h^t \right]}{\delta^2}} + \sqrt{\nu}\|\theta_h^*\|_{\mathcal{H}_k} \Big). \tag{29}$$

Now, $\|\widehat{\theta}_h^t - \theta_h^*\|_{\Lambda_h^t} = \|\Lambda_h^t(\widehat{\theta}_h^t - \theta_h^*)\|_{(\Lambda_h^t)^{-1}}$ dividing both sides by $\|\widehat{\theta}_h^t - \theta_h^*\|_{\Lambda_h^t}$, we get

$$\|\widehat{\theta}_h^t - \theta_h^*\|_{\Lambda_h^t} \leq \Big( \sqrt{\log \frac{\det \left[ (1+\eta) \cdot I + K_h^t \right]}{\delta^2}} + \sqrt{\nu}\|\theta_h^*\|_{\mathcal{H}_k} \Big). \tag{30}$$

Finally, we fix $t = N_1$ and let $\widehat{\theta}_h = \widehat{\theta}_h^t, \Lambda_h = \Lambda_h^t$ and $K_h^{\mathcal{D}_1} = K_h^t$, where $\mathcal{D}_1$ is offline labeled dataset, we get

$$\|\widehat{\theta}_h - \theta_h^*\|_{\Lambda_h} \leq \sqrt{\nu}\|\theta_h^*\|_{\mathcal{H}_k} + \sqrt{\log \frac{\det \left[ \nu I + K_h^{\mathcal{D}_1} \right]}{\delta^2}}. \tag{31}$$

Furthermore, observed that $\det \left( \nu I + K_h^{\mathcal{D}_1} \right) = \det \left( I + \nu^{-1} K_h^{\mathcal{D}_1} \right) \det(\nu I)$. Thus, we have

$$\log \left( \det \left( \nu I + K_h^{\mathcal{D}_1} \right) \right) = \log \left( \det \left( I + \nu^{-1} K_h^{\mathcal{D}_1} \right) \right) + N_1 \log \nu \leq 2G(N_1, \nu) + N_1(\nu - 1), \tag{32}$$

where $\nu > 1$. Thus, set $\nu = 1 + 1/N_1$, we have

$$\|\widehat{\theta}_h - \theta_h^*\|_{\Lambda_h} \leq \sqrt{\nu}\|\theta_h^*\|_{\mathcal{H}_k} + \sqrt{2G(N_1, 1 + 1/N_1) + 1 + \log \frac{1}{\delta^2}}. \tag{33}$$

## B.2 Proof of Theorem 4.3

The proof of Theorem 4.3 and the related supporting lemmas are given in this part. Recall that we denote the labeled dataset $\mathcal{D}_1$, unlabeled dataset $\mathcal{D}_2^{\widetilde{\theta}}$, and $\mathcal{D}^{\widetilde{\theta}} = \{(s_h^\tau, a_h^\tau, \widetilde{r}_h^{\theta_h}(s_h^\tau, a_h^\tau))\}_{\tau, h=1}^{N, H}$, which is a combination of labeled dataset $\mathcal{D}_1$ and unlabeled dataset $\mathcal{D}_2^{\widetilde{\theta}}$ and $N = N_1 + N_2$. We partition dataset $\mathcal{D}^\theta$ into $H$ disjoint and equally sized sub dataset $\{\widetilde{\mathcal{D}}_h^\theta\}_{h=1}^H$, where $|\widetilde{\mathcal{D}}_h^\theta| = N_H = N/H$. Let $\mathcal{I}_h = \{N_H \cdot (h-1) + 1, \ldots, N_H \cdot h\} = \{\tau_{h,1}, \cdots, \tau_{h,N_H}\}$ satisfy $\widetilde{\mathcal{D}}_h^\theta = \{(s_h^\tau, a_h^\tau, \widehat{r}_h^{\theta_h}(s_h^\tau, a_h^\tau))\}_{\tau \in \mathcal{I}_h}$. Denote $\{\widehat{V}_h^{\widetilde{\mathcal{D}}_{h:H}^\theta}\}_{h=1}^H$ as the estimated value function constructed by PEVI. These estimations are based on datasets $\widehat{\mathcal{D}}_{h:H}$, where rewards have been relabeled using the parameter $\theta := \{\theta_h\}_{h=1}^H$. Furthermore, let $V_1^{\pi, \theta}$ be the value function associated with policy $\pi$ and the estimated reward function with parameter $\theta$. From equation (4) in Algorithm 1, for all $h \in [H]$, we have

$$\widehat{V}_h^{\widetilde{\mathcal{D}}_{h:H}^{\widetilde{\theta}}}(s_0) \leq \widehat{V}_h^{\widetilde{\mathcal{D}}_{h:H}^\theta}(s_0), \quad \forall \theta_h \in \mathcal{C}_h(\delta), \tag{34}$$

where $s_0$ is an unique initial state and $\widetilde{\theta} := \{\widetilde{\theta}_h\}_{h=1}^H$ is the pessimistic estimation of $\theta$.

Let $\mathcal{E}_1$ be the event $\theta_h^* \in \mathcal{C}_h(\delta)$, then we have $\mathbb{P}(\mathcal{E}_1) \geq 1 - \delta$ from Proposition 4.1.

Denote $\{\widehat{V}_h^{\widetilde{\mathcal{D}}_{h:H}^{\theta}}\} = \text{Pess}(\mathcal{D})$ from algorithm 3. Let $\mathcal{E}_2$ be the event where the following inequality holds for each dataset $\widetilde{\mathcal{D}}_h^{\theta}$ and

$$\left| \left( \widehat{\mathbb{B}}_h \widehat{V}_{h+1}^{\widetilde{\mathcal{D}}_{h+1:H}^{\theta}} \right)(s,a) - \left( \mathbb{B}_h \widehat{V}_{h+1}^{\widetilde{\mathcal{D}}_{h+1:H}^{\theta}} \right)(s,a) \right| \leq \Gamma_h(s,a), \forall (s,a) \in \mathcal{S} \times \mathcal{A}, h \in [H], \tag{35}$$

where $\Gamma_h(s,a) = B \|\phi(s,a)\|_{(\Lambda_h^{\widetilde{\mathcal{D}}_h^{\theta}})^{-1}}$, and

$$B = \begin{cases} C_2 \cdot H \cdot \sqrt{d \log(N/\delta)} & d\text{-finite spectrum,} \\ C_2 \cdot H \cdot \sqrt{(\log N/\delta)^{1+1/d}} & d\text{-exponential decay,} \\ C_2 \cdot N^{\frac{d+1}{2(d+m)}} H^{1 - \frac{d+1}{2(d+m)}} \cdot \sqrt{\log(N/\delta)} & d\text{-polynomial decay,} \end{cases} \tag{36}$$

where $C_1 > 0$ does not depend on $N_1$ nor $H$ and $C_2 > 0$ does not depend on $N$ nor $H$. Then, apply Lemma B.3 such that $\mathbb{P}(\mathcal{E}_2) \geq 1 - \delta$.

Condition on $\mathcal{E}_1 \cap \mathcal{E}_2$, we have

$$\begin{aligned}
V_1^{\pi^*,\theta^*}(s_0) - V_1^{\widehat{\pi},\theta^*}(s_0) &= V_1^{\pi^*,\theta^*}(s_0) - \widehat{V}_1^{\widetilde{\mathcal{D}}_{1:H}^{\theta^*}}(s_0) + \widehat{V}_1^{\widetilde{\mathcal{D}}_{1:H}^{\theta^*}}(s_0) - V_1^{\widehat{\pi},\theta^*}(s_0) \\
&\leq V_1^{\pi^*,\theta^*}(s_0) - \widehat{V}_1^{\widetilde{\mathcal{D}}_{1:H}^{\theta^*}}(s_0) \\
&= V_1^{\pi^*,\theta^*}(s_0) - V_1^{\pi^*,\widetilde{\theta}}(s_0) + V_1^{\pi^*,\widetilde{\theta}}(s_0) - \widehat{V}_1^{\widetilde{\mathcal{D}}_{1:H}^{\theta}}(s_0) + \widehat{V}_1^{\widetilde{\mathcal{D}}_{1:H}^{\theta}}(s_0) - \widehat{V}_1^{\widetilde{\mathcal{D}}_{1:H}^{\theta^*}}(s_0) \\
&\leq V_1^{\pi^*,\theta^*}(s_0) - V_1^{\pi^*,\widetilde{\theta}}(s_0) + V_1^{\pi^*,\widetilde{\theta}}(s_0) - \widehat{V}_1^{\widetilde{\mathcal{D}}_{1:H}^{\theta}}(s_0) \\
&= V_1^{\pi^*,\theta^*}(s_0) - V_1^{\pi^*,\widehat{\theta}}(s_0) + V_1^{\pi^*,\widehat{\theta}}(s_0) - V_1^{\pi^*,\widetilde{\theta}}(s_0) + V_1^{\pi^*,\widetilde{\theta}}(s_0) - \widehat{V}_1^{\widetilde{\mathcal{D}}_{1:H}^{\theta}}(s_0) \\
&\leq \left| V_1^{\pi^*,\theta^*}(s_0) - V_1^{\pi^*,\widehat{\theta}}(s_0) \right| + \left| V_1^{\pi^*,\widehat{\theta}}(s_0) - V_1^{\pi^*,\widetilde{\theta}}(s_0) \right| + V_1^{\pi^*,\widetilde{\theta}}(s_0) - \widehat{V}_1^{\widetilde{\mathcal{D}}_{1:H}^{\theta}}(s_0) \\
&\leq 2\sum_{h=1}^H \beta_h(\delta) \mathbb{E}_{\pi^*}\left[ \|\phi(s_h,a_h)\|_{(\Lambda_h^{\mathcal{D}_1})^{-1}} \mid s_1 = s_0 \right] \\
&\quad + 2B \sum_{h=1}^H \mathbb{E}_{\pi^*}\left[ \|\phi(s_h,a_h)\|_{(\Lambda_h^{\widetilde{\mathcal{D}}_h^{\theta}})^{-1}} \mid s_1 = s_0 \right],
\end{aligned} \tag{37}$$

for $s_0$ is the unique initial state and $\beta_h(\delta)$ is

$$\beta_h(\delta) = \begin{cases} \sqrt{1 + \frac{1}{N_1}}\mathcal{S} + \sqrt{C_1 \cdot d \cdot \log N_1 + \log(\frac{1}{\delta^2})} & d\text{-finite spectrum,} \\ \sqrt{1 + \frac{1}{N_1}}\mathcal{S} + \sqrt{C_1 \cdot (\log N_1)^{1+\frac{1}{d}} + \log(\frac{1}{\delta^2})} & d\text{-exponential decay,} \\ \sqrt{1 + \frac{1}{N_1}}\mathcal{S} + \sqrt{C_1 \cdot (N_1)^{\frac{d+1}{d+m}} \cdot \log(N_1) + \log(\frac{1}{\delta^2})} & d\text{-polynomial decay.} \end{cases}$$

Note that the first inequality follows Lemma B.1, while the second inequality follows directly from equation (34), and the last inequality follows Lemma B.2 and Theorem B.4.

**Lemma B.1.** *Under the event $\mathcal{E}_2$, we have*

$$\widehat{V}_1^{\widetilde{\mathcal{D}}_{1:H}^{\theta^*}}(s_0) - V_1^{\widehat{\pi},\theta^*}(s_0) \leq 0,$$

*where $s_0$ is the unique initial state.*

*Proof.* For simplicity, we denote $\widehat{V}_1^{\mathcal{D}}(s_0) = \widehat{V}_1^{\widetilde{\mathcal{D}}_{1:H}^{\theta^*}}(s_0)$ and $V_1^{\widehat{\pi}}(s_0) = V_1^{\widehat{\pi},\theta^*}(s_0)$. By Lemma C.3 with $\pi = \pi' = \widehat{\pi}$ and $\{\widehat{Q}_h^{\mathcal{D}}\}_{h=1}^H$ is constructed by PEVI (Algorithm 3), we have

$$\widehat{V}_1^{\mathcal{D}}(s_0) - V_1^{\widehat{\pi}}(s_0) = \sum_{h=1}^H \mathbb{E}_{\widehat{\pi}}\left[\widehat{Q}_h^{\mathcal{D}}(s_h,a_h) - \left(\mathbb{B}_h \widehat{V}_{h+1}^{\mathcal{D}}\right)(s_h,a_h) \mid s_1 = s_0\right].$$

Recall that $\bar{Q}_h^{\mathcal{D}}(s,a)$ is defined in line 6 in Algorithm 3. For all $h \in [H]$ and all $(s,a) \in \mathcal{S} \times \mathcal{A}$, if $\bar{Q}_h^{\mathcal{D}}(s,a) < 0$, implies $\widehat{Q}_h^{\mathcal{D}}(s,a) = 0$. Then

$$\widehat{Q}_h^{\mathcal{D}}(s,a) - \left(\mathbb{B}_h \widehat{V}_{h+1}^{\mathcal{D}}\right)(s,a) = -\left(\mathbb{B}_h \widehat{V}_{h+1}^{\mathcal{D}}\right)(s,a) < 0,$$

as $r_h \in [0,1]$. Otherwise, $\bar{Q}_h^{\mathcal{D}}(s,a) \geq 0$, we have

$$\widehat{Q}_h^{\mathcal{D}}(s,a) - \left(\mathbb{B}_h \widehat{V}_{h+1}^{\mathcal{D}}\right)(s,a) \leq \bar{Q}_h^{\mathcal{D}}(s,a) - \left(\mathbb{B}_h \widehat{V}_{h+1}^{\mathcal{D}}\right)(s,a)$$
$$= \left(\widehat{\mathbb{B}}_h \widehat{V}_{h+1}^{\mathcal{D}}\right)(s,a) - \left(\mathbb{B}_h \widehat{V}_{h+1}^{\mathcal{D}}\right)(s.a) - \Gamma_h(s,a) \leq 0.$$

Finally, we have

$$\widehat{V}_1^{\mathcal{D}}(s_0) - V_1^{\widehat{\pi}}(s_0) \leq 0. \tag{38}$$

$\square$

**Lemma B.2.** *For policy $\pi^*$, and offline dataset $\mathcal{D} = \{(s_h^\tau, a_h^\tau, r_h^\tau)\}_{\tau,h=1}^{N,H}$, and any reward function parameter $\theta_h \in C_h(\delta)$, we have*

$$|V_1^{\pi^*,\theta}(s_0) - V_1^{\pi^*,\widehat{\theta}}(s_0)| \leq \sum_{h=1}^H \beta_h(\delta) \mathbb{E}_{\pi^*}\left[\|\phi(s_h,a_h)\|_{(\Lambda_h^{\mathcal{D}})^{-1}} \mid s_1 = s_0\right], \tag{39}$$

*where $s_0$ is the unique initial state and $\beta_h(\delta)$ is defined in equation (7). Moreover, suppose Assumption 3.1 holds, then $\beta_h(\delta)$ can be written as*

$$\beta_h(\delta) = \begin{cases} \sqrt{1 + \frac{1}{N}}\mathcal{S} + \sqrt{C_1 \cdot d \cdot \log N + 1 + \log(\frac{1}{\delta^2})} & \text{d-finite spectrum,} \\ \sqrt{1 + \frac{1}{N}}\mathcal{S} + \sqrt{C_1 \cdot (\log N)^{1+\frac{1}{d}} + 1 + \log(\frac{1}{\delta^2})} & \text{d-exponential decay,} \\ \sqrt{1 + \frac{1}{N}}\mathcal{S} + \sqrt{C_1 \cdot (N)^{\frac{m+1}{d+m}} \cdot \log(N) + 1 + \log(\frac{1}{\delta^2})} & \text{d-polynomial decay.} \end{cases} \tag{40}$$

*Proof.* From the definition, we have

$$|\widehat{r}_h^{\theta_h}(s,a) - \widehat{r}_h^{\widehat{\theta}_h}(s,a)| = \left|\phi(s,a)^\top \theta_h - \phi(s,a)^\top \widehat{\theta}_h\right|$$
$$\leq \|\theta - \widehat{\theta}_h\|_{\Lambda_h^{\mathcal{D}}} \cdot \|\phi(s,a)\|_{(\Lambda_h^{\mathcal{D}})^{-1}} \tag{41}$$
$$\leq \beta_h(\delta)\sqrt{\phi(s,a)^\top (\Lambda_h^{\mathcal{D}})^{-1}\phi(s,a)}.$$

Then, we have

$$V_1^{\pi^*,\theta}(s_0) - V_1^{\pi^*,\widehat{\theta}}(s_0) = \sum_{h=1}^H \mathbb{E}_{\pi^*}\left[\widehat{r}_h^{\theta_h}(s,a) - \widehat{r}_h^{\widehat{\theta}_h}(s,a) \mid s_1 = s_0\right]$$
$$\leq \sum_{h=1}^H \mathbb{E}_{\pi^*}\left[\left|\widehat{r}_h^{\theta_h}(s,a) - \widehat{r}_h^{\widehat{\theta}_h}(s,a)\right| \mid s_1 = s_0\right]$$
$$\leq \sum_{h=1}^H \beta_h(\delta)\mathbb{E}_{\pi^*}\left[\sqrt{\phi(s_h,a_h)^\top (\Lambda_h^{\mathcal{D}})^{-1}\phi(s_h,a_h)} \mid s_1 = s_0\right] \tag{42}$$
$$= \sum_{h=1}^H \beta_h(\delta)\mathbb{E}_{\pi^*}\left[\|\phi(s_h,a_h)\|_{(\Lambda_h^{\mathcal{D}})^{-1}} \mid s_1 = s_0\right].$$

Recall that $\beta_h(\delta) = \sqrt{\nu}\mathcal{S} + \sqrt{2G(N,\nu) + 1 + \log\frac{1}{\delta^2}}$ defined in equation (7). Setting $\nu = 1 + 1/N$ and applying Lemma C.2, we obtain

$$\beta_h(\delta) = \begin{cases} \sqrt{1 + \frac{1}{N}}\mathcal{S} + \sqrt{C_1 \cdot d \cdot \log N + \log(\frac{1}{\delta^2})} & d\text{-finite spectrum,} \\ \sqrt{1 + \frac{1}{N}}\mathcal{S} + \sqrt{C_1 \cdot (\log N)^{1+\frac{1}{d}} + \log(\frac{1}{\delta^2})} & d\text{-exponential decay,} \\ \sqrt{1 + \frac{1}{N}}\mathcal{S} + \sqrt{C_1 \cdot (N)^{\frac{m+1}{d+m}} \cdot \log(N) + \log(\frac{1}{\delta^2})} & d\text{-polynomial decay,} \end{cases} \tag{43}$$

for some sufficient large $C_1$ and $C_1$ is an absolute constant that does not depend on $N_1$ nor $H$. $\qquad\square$

**Lemma B.3.** *Suppose Assumption 3.2 and 3.1 hold, with dataset $\mathcal{D} = \{(s_h^\tau, a_h^\tau, r_h^\tau)\}_{h,\tau=1}^{H,N}$, we set $\lambda = 1 + 1/N$ and $B > 0$ satisfies*

$$2(1 + \frac{1}{N})R_Q^2 + 8G(N/H, 1 + 1/N) + 2/H + 8\log(H/\xi) \leq (B/H)^2, \tag{44}$$

*in Algorithm 3. Then $\Gamma_h(s,a) = B \cdot \|\phi(s,a)\|_{(\Lambda_h^{\widetilde{\mathcal{D}}_h})^{-1}}$ is a $\xi$-quantifier where $\widetilde{\mathcal{D}}_h$ is defined in Theorem 4.3. That is, for dataset $\mathcal{D}$, the following inequality holds,*

$$\left| \left( \widehat{\mathbb{B}}_h \widehat{V}_{h+1}^{\widetilde{\mathcal{D}}_{h+1:H}} \right)(s,a) - \left( \mathbb{B}_h \widehat{V}_{h+1}^{\widetilde{\mathcal{D}}_{h+1:H}} \right)(s,a) \right| \leq \Gamma_h(s,a) \;, \forall(s,a) \in \mathcal{S} \times \mathcal{A}, h \in [H], \tag{45}$$

*with $\mathbb{P}(\mathcal{E}_2) \geq 1 - \xi$, where $\mathcal{E}_2$ is defined in equation(35). In particular, $B$ is given by*

$$B = \begin{cases} C \cdot H \cdot \sqrt{d \log(N/\xi)} & d\text{-finite spectrum,} \\ C \cdot H \cdot \sqrt{(\log N/\xi)^{1+1/d}} & d\text{-exponential decay,} \\ C \cdot N^{\frac{m+1}{2(d+m)}} H^{1-\frac{m+1}{2(d+m)}} \cdot \sqrt{\log(N/\xi)} & d\text{-polynomial decay,} \end{cases} \tag{46}$$

*for some absolute constant $C$ that does not depend on $N$ nor $H$.*

*Proof.* We present the offline reinforcement setting of (Yang et al., 2020) and combine the data split skill from (Xie et al., 2021) (Rashidinejad et al., 2021). Recall that we partition dataset $\mathcal{D}$ into $H$ disjoint and equally sized sub dataset $\{\widetilde{\mathcal{D}}_h\}_{h=1}^H$, where $|\widetilde{\mathcal{D}}_h| = N_H = N/H$. Let $\mathcal{I}_h = \{N_H \cdot (h-1) + 1, \ldots, N_H \cdot h\} = \{\tau_{h,1}, \cdots, \tau_{h,N_H}\}$ satisfy $\widetilde{\mathcal{D}}_h = \{(s_h^\tau, a_h^\tau, r_h^\tau)\}_{\tau \in \mathcal{I}_h}$. Denote the operator $\Phi_h^{\widetilde{\mathcal{D}}_h} : \mathcal{H}_k \to \mathbb{R}^{N_H}$ and $\Lambda_h^{\widetilde{\mathcal{D}}_h} : \mathcal{H}_k \to \mathcal{H}_k$ as

$$\Phi_h^{\widetilde{\mathcal{D}}_h} = \begin{pmatrix} \phi\left(z_h^{\tau_{h,1}}\right)^\top \\ \vdots \\ \phi\left(z_h^{\tau_{h,N_H}}\right)^\top \end{pmatrix} = \begin{pmatrix} k\left(\cdot, z_h^{\tau_{h,1}}\right)^\top \\ \vdots \\ k\left(\cdot, z_h^{\tau_{h,N_H}}\right)^\top \end{pmatrix}, \quad \Lambda_h^{\widetilde{\mathcal{D}}_h} = \lambda \cdot I_{\mathcal{H}} + (\Phi_h^{\widetilde{\mathcal{D}}_h})^\top \Phi_h^{\widetilde{\mathcal{D}}_h}. \tag{47}$$

For notation simplicity, let $K_h = K_h^{\widetilde{\mathcal{D}}_h}$, $k_h(z) = k_h^{\widetilde{\mathcal{D}}_h}(z)$, and $\Lambda_h = \Lambda_h^{\widetilde{\mathcal{D}}_h}$. For $h \in [H]$, the solution $\widehat{f}_h$ of the kernel ridge regression in equation (15) is that

$$\widehat{f}_h(\cdot) = \sum_{\tau \in \mathcal{I}_h} [\widehat{\alpha}_h]_\tau k(z_h^\tau, \cdot) = \Phi_h^\top \widehat{\alpha}_h, \tag{48}$$

where $\widehat{\alpha}_h = (K_h + \lambda \cdot I)^{-1} y_h$. Note that both matrix $\Phi_h \Phi_h^\top + \lambda \cdot I$ and $\Phi_h^\top \Phi_h + \lambda \cdot I_{\mathcal{H}}$ are positive definite, following (Valko et al., 2013), we have

$$\Phi_h^\top \left( \Phi_h \Phi_h^\top + \lambda \cdot I \right)^{-1} = \left( \Phi_h^\top \Phi_h + \lambda \cdot I_{\mathcal{H}} \right)^{-1} \Phi_h^\top. \tag{49}$$

Then, the fitted value function $\widehat{\mathbb{B}}_h \widehat{V}_{h+1}^{\widetilde{\mathcal{D}}_{h+1:H}}$ is that

$$\widehat{f}_h(z) = \langle \widehat{f}_h, k(\cdot, z) \rangle_{\mathcal{H}_k} = k_h(z)^\top \widehat{\alpha}_h. \tag{50}$$

Furthermore, in combination with the equation (49), $k_h(z)$ can be written as $k_h(z) = \Phi_h \phi(z)$, we have

$$
\begin{aligned}
\phi(z) &= \Lambda_h^{-1} \Lambda_h \phi(z) \\
&= \Lambda_h^{-1} \left[ \Phi_h^\top \Phi_h + \lambda \cdot I_{\mathcal{H}_k} \right] \phi(z) \\
&= \Lambda_h^{-1} \Phi_h^\top \Phi_h \phi(z) + \lambda \Lambda_h^{-1} \phi(z) \\
&= \Phi_h^\top (K_h + \lambda \cdot I)^{-1} \Phi_h \phi(z) + \lambda \Lambda_h^{-1} \phi(z) \\
&= \Phi_h^\top (K_h + \lambda \cdot I)^{-1} k_h(z) + \lambda \Lambda_h^{-1} \phi(z),
\end{aligned}
\tag{51}
$$

where the forth equality follows equation (49). Recall that $\widehat{Q}_h^{\widetilde{\mathcal{D}}_{h:H}}(z) = \min\{\bar{Q}_h^{\widetilde{\mathcal{D}}_{h:H}}(z), H - h + 1\}^+ = \min\{k_h(z)^\top \widehat{\alpha}_h - \Gamma_h(z), H - h + 1\}^+$ in Algorithm 3. Since $\widehat{Q}_{h+1}^{\widetilde{\mathcal{D}}_{h+1:H}} \in [0, H]$ for all $h \in [H]$, by Assumption 3.2, we have $\mathbb{B}_h \widehat{V}_{h+1}^{\widetilde{\mathcal{D}}_{h+1:H}} \in \mathcal{Q}^*$, i.e., $\left| \mathbb{B}_h \widehat{V}_{h+1}^{\widetilde{\mathcal{D}}_{h+1:H}} \right|_{\mathcal{H}} \le R_Q H$. There exists $f_h \in \mathcal{Q}^*$ such that $f_h = \mathbb{B}_h \widehat{V}_{h+1}^{\widetilde{\mathcal{D}}_{h+1:H}}$ and $\mathbb{B}_h \widehat{V}_{h+1}^{\widetilde{\mathcal{D}}_{h+1:H}}(z) = \langle f_h, k(\cdot, z) \rangle_{\mathcal{H}_k} = \phi(z)^\top f_h$ by the feature representation of RKHS. For any $h \in [H]$,

$$
\begin{aligned}
&\left| \left( \widehat{\mathbb{B}}_h \widehat{V}_{h+1}^{\widetilde{\mathcal{D}}_{h+1:H}} \right)(s, a) - \left( \mathbb{B}_h \widehat{V}_{h+1}^{\widetilde{\mathcal{D}}_{h+1:H}} \right)(s, a) \right| \\
&= \left| \widehat{f}_h(s, a) - \phi(s, a)^\top f_h \right| \\
&= \left| k_h(z)^\top (K_h + \lambda \cdot I)^{-1} y_h - k_h^\top(z)(K_h + \lambda \cdot I)^{-1} \Phi_h f_h - \lambda \phi(z)^\top \Lambda_h^{-1} f_h \right| \\
&\le \left| k_h(z)^\top (K_h + \lambda \cdot I)^{-1} (y_h - \Phi_h f_h) \right| + \left| \lambda \phi(z)^\top \Lambda_h^{-1} f_h \right| \\
&\le (A) + (B),
\end{aligned}
\tag{52}
$$

where the second equality follows equation (51). Next, we bound $(A)$ and $(B)$ separately. By Cauchy-Schwarz inequality,

$$
\begin{aligned}
(B) &= \left| \lambda \phi(z)^\top \Lambda_h^{-1} f_h \right| = \lambda \langle \Lambda_h^{-1} \phi(z), f_h \rangle_{\mathcal{H}_k} \\
&\le \lambda \cdot \left\| \Lambda_h^{-1} \phi(z) \right\|_{\mathcal{H}_k} \cdot \| f_h \|_{\mathcal{H}_k} \\
&= \lambda \cdot \left\| \Lambda_h^{-1/2} \Lambda_h^{-1/2} \phi(z) \right\|_{\mathcal{H}_k} \cdot \| f_h \|_{\mathcal{H}_k} \\
&\le R_Q H \cdot \lambda^{1/2} \cdot \left\| \Lambda_h^{-1/2} \phi(z) \right\|_{\mathcal{H}_k} \\
&= R_Q H \cdot \lambda^{1/2} \cdot \| \phi(z) \|_{\Lambda_h^{-1}}.
\end{aligned}
\tag{53}
$$

Furthermore, $y_h$ is defined in equation (18) and Section 3.1, the $\tau$-th entry of $(y_h - \Phi_h f_h)$ can be written as

$$
\begin{aligned}
[y_h]_\tau - [\Phi_h f_h]_\tau &= r_h^\tau + \widehat{V}_{h+1}^{\widetilde{\mathcal{D}}_{h+1:H}}\left(s_{h+1}^\tau\right) - \phi(s_h^\tau, a_h^\tau) f_h \\
&= r_h^\tau + \widehat{V}_{h+1}^{\widetilde{\mathcal{D}}_{h+1:H}}\left(s_{h+1}^\tau\right) - \mathbb{B}_h \widehat{V}_{h+1}^{\widetilde{\mathcal{D}}_{h+1:H}}\left(s_h^\tau, a_h^\tau\right) \\
&= \widehat{V}_{h+1}^{\widetilde{\mathcal{D}}_{h+1:H}}\left(s_{h+1}^\tau\right) - \mathbb{P}_h \widehat{V}_{h+1}^{\widetilde{\mathcal{D}}_{h+1:H}}\left(s_h^\tau, a_h^\tau\right) + \epsilon_h^\tau.
\end{aligned}
\tag{54}
$$

With equation (49) and $k_h(z) = \Phi_h \phi(z)$, $(A)$ can be written as

$$
\begin{aligned}
(A) &= \left| k_h(z)^\top (K_h + \lambda \cdot I)^{-1} (y_h - \Phi_h f_h) \right| \\
&= \left| \phi(s, a)^\top \Phi_h^\top (K_h + \lambda \cdot I)^{-1} (y_h - \Phi_h f_h) \right| \\
&= \left| \phi(s, a)^\top \Lambda_h^{-1} \Phi_h^\top (y_h - \Phi_h f_h) \right| \\
&= \left| \phi(s, a)^\top \Lambda_h^{-1} \sum_{\tau \in \mathcal{I}_h} \phi\left(s_h^\tau, a_h^\tau\right) \left[ \widehat{V}_{h+1}^{\widetilde{\mathcal{D}}_{h+1:H}}\left(s_{h+1}^\tau\right) - \mathbb{P}_h \widehat{V}_{h+1}^{\widetilde{\mathcal{D}}_{h+1:H}}\left(s_h^\tau, a_h^\tau\right) + \epsilon_h^\tau \right] \right| \\
&\le \| \phi(s, a) \|_{\Lambda_h^{-1}} \cdot \left\| \sum_{\tau \in \mathcal{I}_h} \phi\left(s_h^\tau, a_h^\tau\right) \left[ \widehat{V}_{h+1}^{\widetilde{\mathcal{D}}_{h+1:H}}\left(s_{h+1}^\tau\right) - \mathbb{P}_h \widehat{V}_{h+1}^{\widetilde{\mathcal{D}}_{h+1:H}}\left(s_h^\tau, a_h^\tau\right) + \epsilon_h^\tau \right] \right\|_{\Lambda_h^{-1}},
\end{aligned}
\tag{55}
$$

where the last inequality uses Cauchy-Schwarz inequality.

For $h \in [H-1]$, we define the filtration

$$\mathcal{F}_h = \sigma\left(\widetilde{\mathcal{D}}_1 \cup \cdots \cup \widetilde{\mathcal{D}}_h\right),$$

where $\sigma(\cdot)$ is the $\sigma$-algebra generated by the set of random variables. Let

$$\varepsilon_h^\tau = (\widehat{V}_{h+1}^{\widetilde{\mathcal{D}}_{h+1:H}}\left(s_{h+1}^\tau\right) + \epsilon_h^\tau) - \mathbb{P}_h \widehat{V}_{h+1}^{\widetilde{\mathcal{D}}_{h+1:H}}\left(s_h^\tau, a_h^\tau\right),$$

is adapted to the filtration $\{\mathcal{F}_{h+1}\}_{h=1}^{H-1}$. Then

$$\mathbb{E}\left[\widehat{V}_{h+1}^{\widetilde{\mathcal{D}}_{h+1:H}}\left(s_{h+1}^\tau\right) + \epsilon_h^\tau \mid \mathcal{F}_h\right] = \mathbb{E}\left[\widehat{V}_{h+1}^{\widetilde{\mathcal{D}}_{h+1:H}}\left(s_{h+1}\right) + \epsilon_h^\tau \mid s_h = s_h^\tau, a_h = a_h^\tau\right] = \mathbb{P}_h \widehat{V}_{h+1}^{\widetilde{\mathcal{D}}_{h+1:H}}\left(s_h^\tau, a_h^\tau\right).$$

Thus, we have $\mathbb{E}\left[\varepsilon_h^\tau \mid \mathcal{F}_h\right] = 0$. Applying Lemma C.1 to $\epsilon_\tau = \varepsilon_h^\tau$ and $\sigma^2 = 2H^2$ as

$$\widehat{V}_{h+1}^{\widetilde{\mathcal{D}}_{h+1:H}}\left(s_{h+1}^\tau\right) - \mathbb{P}_h \widehat{V}_{h+1}^{\widetilde{\mathcal{D}}_{h+1:H}}\left(s_h^\tau, a_h^\tau\right) \in [-H, H], \tag{56}$$

and $\epsilon_h^\tau$ is 1-sub Gaussian noise, for any $\eta > 0$ and $\xi > 0$, it holds probability at least $1 - \xi/H$ that

$$\begin{aligned} & E_h^\top \left[(K_h + \eta \cdot I)^{-1} + I\right]^{-1} E_h \\ & \leq 2H^2 \cdot \log\det\left[(1+\eta) \cdot I + K_h\right] + 4H^2 \cdot \log(H/\xi). \end{aligned} \tag{57}$$

where $E_h = \begin{pmatrix} \varepsilon_h^{\tau_{h,1}} \\ \vdots \\ \varepsilon_h^{\tau_{h,N_H}} \end{pmatrix}$. Using the equation (57), we get

$$\begin{aligned} \left\|\sum_{\tau \in \mathcal{I}_h} \phi\left(s_h^\tau, a_h^\tau\right) \varepsilon_h^\tau\right\|_{\Lambda_h^{-1}}^2 &= E_h^\top \Phi_h \left(\Phi_h^\top \Phi_h + \lambda \cdot I_{\mathcal{H}}\right)^{-1} \Phi_h^\top E_h \\ &= E_h^\top \Phi_h \Phi_h^\top \left(\Phi_h \Phi_h^\top + \lambda \cdot I\right)^{-1} E_h \\ &= E_h^\top K_h \left(K_h + \lambda \cdot I\right)^{-1} E_h. \end{aligned} \tag{58}$$

For $\lambda = \eta + 1 > 1$ and $\eta > 0$, we have

$$\begin{aligned} E_h^\top K_h \left(K_h + \lambda \cdot I\right)^{-1} E_h &= E_h^\top K_h \left(K_h + (\eta+1) \cdot I\right)^{-1} E_h \\ &\leq E_h^\top \left(K_h + \eta \cdot I\right) \left(K_h + (\eta+1) \cdot I\right)^{-1} E_h \\ &= E_h^\top \left[(K_h + \eta \cdot I)^{-1} + I\right]^{-1} E_h, \end{aligned} \tag{59}$$

where the first equality follows the fact $\left((K_h + \eta \cdot I)^{-1} + I\right)^{-1} = (K_h + \eta \cdot I)(K_h + (1+\eta) \cdot I)^{-1}$. For any fixed $\xi > 0$, combining equation (57), (58), and (59), we get

$$\left\|\sum_{\tau \in \mathcal{I}_h} \phi\left(s_h^\tau, a_h^\tau\right) \varepsilon_h^\tau\right\|_{\Lambda_h^{-1}}^2 \leq 2H^2 \cdot \log\det\left[\lambda \cdot I + K_h\right] + 4H^2 \cdot \log(H/\xi), \tag{60}$$

holds simultaneously for all $h \in [H]$ with probability at least $1 - \xi$. Clearly, $\lambda \cdot I + K_h = (\lambda \cdot I)(I + K_h/\lambda)$, then

$$\begin{aligned} \log\det\left(\lambda \cdot I + K_h\right) &= N_H \log\lambda + \log\det\left(I + K_h/\lambda\right) \\ &\leq N_H(\lambda - 1) + \log\det\left(I + K_h/\lambda\right), \end{aligned} \tag{61}$$

where $N_H = |\mathcal{I}_h| = N/H$. Hence, for any $\varepsilon > 0$ and $\lambda > 1$,

$$\left\| \sum_{\tau \in \mathcal{I}_h} \phi\left(s_h^\tau, a_h^\tau\right) \varepsilon_h^\tau \right\|_{\Lambda_h^{-1}}^2 \leq 2H^2 \cdot \log \det \left[I + K_h/\lambda\right] + H^2 N_H(\lambda - 1) + 4H^2 \cdot \log(H/\xi), \tag{62}$$

holds simultaneously for all $h \in [H]$ with probability at least $1 - \xi$. Finally, combine equation (52), (53), (55), and (62) and take $\lambda = 1 + \frac{1}{N}$, we get

$$\begin{aligned}
&\left| \left(\widehat{\mathbb{B}}_h \widehat{V}_{h+1}^{\widetilde{\mathcal{D}}_{h+1:H}}\right)(z) - \left(\mathbb{B}_h \widehat{V}_{h+1}^{\widetilde{\mathcal{D}}_{h+1:H}}\right)(z) \right| \\
&\leq \|\phi(z)\|_{\Lambda_h^{-1}} \left[ R_Q H \cdot \sqrt{\lambda} + \left\| \sum_{\tau \in \mathcal{I}_h} \phi\left(s_h^\tau, a_h^\tau\right) \varepsilon_h^\tau \right\|_{\Lambda_h^{-1}} \right] \\
&\leq \|\phi(z)\|_{\Lambda_h^{-1}} \left[ 2\lambda R_Q^2 H^2 + 2 \left\| \sum_{\tau \in \mathcal{I}_h} \phi\left(s_h^\tau, a_h^\tau\right) \varepsilon_h^\tau \right\|_{\Lambda_h^{-1}}^2 \right]^{1/2} \\
&\leq \|\phi(z)\|_{\Lambda_h^{-1}} \left[ 2\lambda R_Q^2 H^2 + 8H^2 \cdot G(N/H, \lambda) + 2H^2 N_H(\lambda - 1) + 8H^2 \cdot \log(H/\xi) \right]^{1/2} \\
&\leq \|\phi(z)\|_{\Lambda_h^{-1}} \left[ 2(1 + \frac{1}{N}) R_Q^2 H^2 + 8H^2 \cdot G(N/H, 1 + 1/N) + 2H + 8H^2 \cdot \log(H/\xi) \right]^{1/2} \\
&\leq B \cdot \|\phi(z)\|_{\Lambda_h^{-1}} = \Gamma_h(z), \quad \forall z \in \mathcal{S} \times \mathcal{A},
\end{aligned} \tag{63}$$

where the second inequality follows from $\sqrt{x} + \sqrt{y} \leq \sqrt{2(x^2 + y^2)}$. Thus, $\{\Gamma_h\}_{h \in [H]}$ is a $\xi$-uncertainty quantifier. To give explicit expressions for $B$, we distinguish three cases according to the spectrum of $k$.

- Case I ($d$-finite spectrum): By Lemma C.2, we have $G(N/H, 1 + 1/N) \leq C_k \cdot d \log(N/H)$, where $C_k$ is absolute constant that depends on $m$, and $d$. Then, $B^2$ equals to

$$\begin{aligned}
&2(1 + \frac{1}{N}) R_Q^2 H^2 + 8H^2 \cdot G(N/H, 1 + 1/N) + 2H + 8H^2 \cdot \log(H/\xi) \\
&\leq 4R_Q^2 H^2 + 8H^2 \cdot C_k \cdot d \log(N/H) + 2H + 8H^2 \cdot \log(H/\xi) \\
&\leq C^2 \cdot H^2 \cdot d \log(N/\xi),
\end{aligned} \tag{64}$$

for sufficient large $C > 0$. Hence, we take $B = C \cdot H \cdot \sqrt{d \log(N/\xi)}$.

- Case II ($d$-exponential decay): By Lemma C.2, we get

$$G(N/H, 1 + 1/N) \leq C_k \cdot (\log(N/H))^{1+1/d}, \tag{65}$$

where $C_k$ is absolute constant that only depends on $m$ and $d$. Then, $B^2$ equals to

$$\begin{aligned}
&2(1 + \frac{1}{N}) R_Q^2 H^2 + 8H^2 \cdot G(N/H, 1 + 1/N) + 2H + 8H^2 \cdot \log(H/\xi) \\
&\leq 4R_Q^2 H^2 + 8H^2 \cdot C_k \cdot (\log(N/H))^{1+1/d} + 2H + 8H^2 \cdot \log(H/\xi) \\
&\leq C^2 H^2 \cdot \left[ (\log(N/H))^{1+1/d} + \log(H/\xi) \right] \\
&\leq C^2 H^2 \cdot [\log(N/H) + \log(H/\xi)]^{1+1/d} \\
&\leq C^2 \cdot H^2 \cdot [\log(N/\xi)]^{1+1/d},
\end{aligned} \tag{66}$$

for sufficient large $C > 0$. Hence, we take $B = C \cdot H \cdot \sqrt{(\log N/\xi)^{1+1/d}}$.

- Case III: ($d$-polynomial decay) By Lemma C.2, we get

$$G(N/H, 1 + 1/N) \leq C_k \cdot (N/H)^{\frac{m+1}{d+m}} \cdot \log(N/H), \tag{67}$$

where $C_k$ is absolute constant that only depends on $m$ and $d$. Then, $B^2$ equals to

$$
\begin{aligned}
&2(1 + \frac{1}{N})R_Q^2 H^2 + 8H^2 \cdot G(N/H, 1 + 1/N) + 2H + 8H^2 \cdot \log(H/\xi) \\
&\leq 4R_Q^2 H^2 + 8H^2 \cdot C_k \cdot (N/H)^{\frac{m+1}{d+m}} \cdot \log(N/H) + 2H + 8H^2 \cdot \log(H/\xi) \\
&\leq C^2 H^{2 - \frac{m+1}{d+m}} N^{\frac{m+1}{d+m}} \cdot [\log(N/H) + \log(H/\xi)] \\
&\leq C^2 H^{2 - \frac{m+1}{d+m}} N^{\frac{m+1}{d+m}} \cdot [\log(N/\xi)],
\end{aligned}
\tag{68}
$$

for sufficient large $C > 0$. Thus, it suffices to choose $B = C \cdot N^{\frac{m+1}{2(d+m)}} H^{1 - \frac{m+1}{2(d+m)}} \cdot \sqrt{\log(N/\xi)}$.

$\square$

**Theorem B.4.** *Under Assumption 3.2 and 3.1, we set $\lambda = 1 + 1/N$ and $B$ is defined as equation (46). Then with probability at least $1 - \xi$, it holds that*

$$V_1^{\pi^*, \widetilde{\theta}}(s_0) - \widehat{V}_1^{\widetilde{\mathcal{D}}_{1:H}^\theta}(s_0) \leq 2B \sum_{h=1}^{H} \mathbb{E}_{\pi^*} \left[ \|\phi(s_h, a_h)\|_{(\Lambda_h^{\widetilde{\mathcal{D}}_h^\theta})^{-1}} \mid s_1 = s_0 \right]. \tag{69}$$

*Proof.* Recall that the $\xi$-quantifier satisfies the following inequality with probability at least $1 - \xi$:

$$\left| \left( \widehat{\mathbb{B}}_h \widehat{V}_{h+1}^{\widetilde{\mathcal{D}}_{h+1:H}^\theta} \right)(s, a) - \left( \mathbb{B}_h \widehat{V}_{h+1}^{\widetilde{\mathcal{D}}_{h+1:H}^\theta} \right)(s, a) \right| \leq \Gamma_h(s, a), \tag{70}$$

for all $(s, a) \in \mathcal{S} \times \mathcal{A}, h \in [H]$. Define $\widehat{\pi} = \{\widehat{\pi}_h\}_{h=1}^H$ as the policy such that $\widehat{V}_h^{\widetilde{\mathcal{D}}_{h:H}^\theta}(s) = \left\langle \widehat{Q}_h^{\widetilde{\mathcal{D}}_{h:H}^\theta}(s, \cdot), \widehat{\pi}_h(\cdot \mid s) \right\rangle_{\mathcal{A}}$. For simplicity, we denote $\delta_h(s, a) = \left( \mathbb{B}_h \widehat{V}_{h+1}^{\widetilde{\mathcal{D}}_{h+1:H}^\theta} \right)(s, a) - \widehat{Q}_h^{\widetilde{\mathcal{D}}_{h:H}^\theta}(s, a)$ Applying Lemma C.3 with $\pi = \widehat{\pi}$, and $\pi' = \pi^*$, we have

$$
\begin{aligned}
\widehat{V}_1^{\widetilde{\mathcal{D}}_{1:H}^\theta}(s_0) - V_1^{\pi^*, \widetilde{\theta}}(s_0) &= \sum_{h=1}^{H} \mathbb{E}_{\pi^*} \left[ \left\langle \widehat{Q}_h^{\widetilde{\mathcal{D}}_{h:H}^\theta}(s_h, \cdot), \widehat{\pi}_h(\cdot \mid s_h) - \pi_h^*(\cdot \mid s_h) \right\rangle_{\mathcal{A}} \mid s_1 = s_0 \right] \\
&+ \sum_{h=1}^{H} \mathbb{E}_{\pi^*} \left[ \widehat{Q}_h^{\widetilde{\mathcal{D}}_{h:H}^\theta}(s_h, a_h) - \left( \mathbb{B}_h \widehat{V}_{h+1}^{\widetilde{\mathcal{D}}_{h+1:H}^\theta} \right)(s_h, a_h) \mid s_1 = s_0 \right] \\
&= \sum_{h=1}^{H} \mathbb{E}_{\pi^*} \left[ \left\langle \widehat{Q}_h^{\widetilde{\mathcal{D}}_{h:H}^\theta}(s_h, \cdot), \widehat{\pi}_h(\cdot \mid s_h) - \pi_h^*(\cdot \mid s_h) \right\rangle_{\mathcal{A}} \mid s_1 = s_0 \right] \\
&- \sum_{h=1}^{H} \mathbb{E}_{\pi^*} \left[ \delta_h(s_h, a_h) \mid s_1 = s_0 \right],
\end{aligned}
\tag{71}
$$

where $\mathbb{E}_{\pi^*}$ is taken with respect to the trajectory generated by $\pi^*$. Since $\widehat{\pi}$ is greedy with respect to $\widehat{Q}_h^{\widetilde{\mathcal{D}}_{h:H}^\theta}$, then

$$
\begin{aligned}
V_1^{\pi^*, \widetilde{\theta}}(s_0) - \widehat{V}_1^{\widetilde{\mathcal{D}}_{1:H}^\theta}(s_0) &= \sum_{h=1}^{H} \mathbb{E}_{\pi^*} \left[ \delta_h(s_h, a_h) \mid s_1 = s_0 \right] \\
&+ \sum_{h=1}^{H} \mathbb{E}_{\pi^*} \left[ \left\langle \widehat{Q}_h^{\widetilde{\mathcal{D}}_{h:H}^\theta}(s_h, \cdot), \pi_h^*(\cdot \mid s_h) - \widehat{\pi}_h(\cdot \mid s_h) \right\rangle_{\mathcal{A}} \mid s_1 = s_0 \right] \\
&\leq \sum_{h=1}^{H} \mathbb{E}_{\pi^*} \left[ \delta_h(s_h, a_h) \mid s_1 = s_0 \right].
\end{aligned}
\tag{72}
$$

Recall that the construction of $\bar{Q}_h^{\widetilde{\mathcal{D}}_{h:H}^{\theta}}$ in Line 5 in Algorithm 2. For all $h \in [H]$ and all $(s,a) \in \mathcal{S} \times \mathcal{A}$, we have

$$
\begin{aligned}
\bar{Q}_h^{\widetilde{\mathcal{D}}_{h:H}^{\theta}}(s,a) &= \widehat{\mathbb{B}}_h \widehat{V}_{h+1}^{\widetilde{\mathcal{D}}_{h+1:H}^{\theta}}(s,a) - \Gamma_h(s,a) \\
&\leq \mathbb{B}_h \widehat{V}_{h+1}^{\widetilde{\mathcal{D}}_{h+1:H}^{\theta}}(s,a) \leq H - h + 1,
\end{aligned}
\tag{73}
$$

where the first inequality follows the definition of $\Gamma_h(s,a)$ and the second inequality follows that $r_h \in [0,1]$ and $\widehat{V}_{h+1}^{\widetilde{\mathcal{D}}_{h+1:H}^{\theta}}(s,a) \leq H - h$. Then, we have

$$
\widehat{Q}_h^{\widetilde{\mathcal{D}}_{h:H}^{\theta}}(s,a) = \max\left\{\bar{Q}_h^{\widetilde{\mathcal{D}}_{h:H}^{\theta}}(s,a), 0\right\} \geq \bar{Q}_h^{\widetilde{\mathcal{D}}_{h:H}^{\theta}}(s,a).
\tag{74}
$$

Then, $\delta_h(s,a)$ can be written as

$$
\begin{aligned}
\delta_h(s,a) &= \left(\mathbb{B}_h \widehat{V}_{h+1}^{\widetilde{\mathcal{D}}_{h+1}^{\theta}}\right)(s,a) - \widehat{Q}_h^{\widetilde{\theta}, \widetilde{\mathcal{D}}_h}(s,a) \\
&\leq \left(\mathbb{B}_h \widehat{V}_{h+1}^{\widetilde{\mathcal{D}}_{h+1}^{\theta}}\right)(s,a) - \bar{Q}_h^{\widetilde{\theta}, \widetilde{\mathcal{D}}_h}(s,a) \\
&\leq \left(\mathbb{B}_h \widehat{V}_{h+1}^{\widetilde{\mathcal{D}}_{h+1}^{\theta}}\right)(s,a) - \left(\widehat{\mathbb{B}}_h \widehat{V}_{h+1}^{\widetilde{\mathcal{D}}_{h+1}^{\theta}}\right)(s,a) + \Gamma_h(s,a) \leq 2\Gamma_h(s,a),
\end{aligned}
\tag{75}
$$

where the inequality follows line 4 in Algorithm 2. Hence we have $\delta_h(s,a) \leq 2\Gamma_h(s,a)$. Combine Lemma B.3, equation (72), and equation (75), we get

$$
\begin{aligned}
V_1^{\pi^*, \widetilde{\theta}}(s_0) - \widehat{V}_1^{\widetilde{\mathcal{D}}_{1:H}^{\theta}}(s_0) &\leq 2\sum_{h=1}^{H} \mathbb{E}_{\pi^*}\left[\Gamma_h(s_h, a_h) \mid s_1 = s_0\right] \\
&\leq 2B \sum_{h=1}^{H} \mathbb{E}_{\pi^*}\left[\|\phi(s_h, a_h)\|_{(\Lambda_h^{\widetilde{\mathcal{D}}_h^{\theta}})^{-1}} \mid s_1 = s_0\right],
\end{aligned}
\tag{76}
$$

where $B$ satisfies equation (44). $\qquad \square$

## B.3 Proof of Proposition 4.5

Denote $\Lambda_h = \Lambda_h^{\widetilde{\mathcal{D}}_h}, \Lambda_h' = \Lambda_h^{\widetilde{\mathcal{D}}_h'}$, and $K_h' = K_h^{\widetilde{\mathcal{D}}_h'}$. Note that $\Lambda_h$ is a self-adjoint operator, then

$$
\Lambda_h' = \Lambda_h^{1/2}\left(I_{\mathcal{H}_k} + \Lambda_h^{-1/2}\phi(z)\phi(z)^\top \Lambda_h^{-1/2}\right)\Lambda_h^{1/2}.
\tag{77}
$$

We take $\log\det$ on both sides with the equation (77), then

$$
\begin{aligned}
\log\det(\Lambda_h') &= \log\det(\Lambda_h) + \log\det\left(I_{\mathcal{H}_k} + \Lambda_h^{-1/2}\phi(z)\phi(z)^\top \Lambda_h^{-1/2}\right) \\
&= \log\det(\Lambda_h) + \log\left(1 + \phi(z)^\top \Lambda_h^{-1}\phi(z)\right).
\end{aligned}
\tag{78}
$$

Note that $\det(\Lambda_h) = \det(\lambda I + K_h)$, and $\det(\Lambda_h') = \det(\lambda I + K_h')$ for $\lambda \geq 1$ because $\phi(z)^\top \Lambda_h^{-1}\phi(z) \leq 1$, we have

$$
\begin{aligned}
\phi(z)^\top \Lambda_h^{-1}\phi(z) &\leq 2\log\left(1 + \phi(z)^\top \Lambda_h^{-1}\phi(z)\right) \\
&= 2 \cdot \left[\log\det(\Lambda_h') - \log\det(\Lambda_h)\right] \\
&= 2 \cdot \left[\log\det(I + K_h'/\lambda) - \log\det(I + K_h/\lambda)\right].
\end{aligned}
$$

Moreover, by equation (2), we have

$$
\phi(z)^\top \Lambda_h^{-1}\phi(z) \leq 2 \cdot \left[\log\det(I + K_h'/\lambda) - \log\det(I + K_h/\lambda)\right].
\tag{79}
$$

### B.4 Proof of Corollary 4.8

The proof is inspired from (Jin et al., 2021; Duan et al., 2020). Recall that we denote $\mathcal{D}^{\widetilde{\theta}} = \{(s_h^\tau, a_h^\tau, \widetilde{r}_h^{\widetilde{\theta}_h}(s_h^\tau, a_h^\tau))\}_{\tau,h=1}^{N,H}$, which is a combination of labeled dataset $\mathcal{D}_1$ and unlabeled dataset $\mathcal{D}_2^{\widetilde{\theta}}$. We partition dataset $\mathcal{D}^{\widetilde{\theta}}$ into $H$ disjoint and equally sized sub dataset $\{\widetilde{\mathcal{D}}_h^{\widetilde{\theta}}\}_{h=1}^H$, where $|\widetilde{\mathcal{D}}_h^{\widetilde{\theta}}| = N_H = N/H$. Let $\mathcal{I}_h = \{N_H \cdot (h-1) + 1, \ldots, N_H \cdot h\} = \{\tau_{h,1}, \cdots, \tau_{h,N_H}\}$ satisfy $\widetilde{\mathcal{D}}_h^{\widetilde{\theta}} = \{(s_h^\tau, a_h^\tau, \widetilde{r}_h^{\widetilde{\theta}_h}(s_h^\tau, a_h^\tau))\}_{\tau \in \mathcal{I}_h}$. Define

$$Z_h = \sum_{\tau \in \mathcal{I}_h} A_h^\tau, \quad A_h^\tau = \phi\left(s_h^\tau, a_h^\tau\right)\phi\left(s_h^\tau, a_h^\tau\right)^\top - \Sigma_h, \tag{80}$$

where $\Sigma_h = \mathbb{E}_{\bar{\pi}}\left[\phi\left(s_h, a_h\right)\phi\left(s_h, a_h\right)^\top\right]$ for all $h \in [H]$. Clearly, $\mathbb{E}_{\bar{\pi}}[A_h^\tau] = 0$ from equation (80). Note that $\mathbb{E}_{\bar{\pi}}$ is taken with respect to the trajectory induced by the fixed behavior policy $\bar{\pi}$ in the underlying MDP, and the set $\{A_h^\tau\}_{\tau \in \mathcal{I}_h}$ is i.i.d. and centered for all $h \in [H]$.

As shown in Section 3.3, the feature mapping $\phi : \mathcal{Z} \to \mathcal{H}$ satisfies

$$\phi(z) = \sum_{j=1}^\infty \sigma_j \cdot \psi_j(z) \cdot \psi_j = \sum_{j=1}^\infty \sqrt{\sigma_j} \cdot \psi_j(z) \cdot \left(\sqrt{\sigma_j} \cdot \psi_j\right). \tag{81}$$

Let $t$ be any positive integer and let $\Pi_t : \mathcal{H} \to \mathcal{H}$ denote the projection onto the subspace spanned by $\{\psi_j\}_{j \in [t]}$, i.e., $\Pi_t[\phi(z)] = \sum_{j=1}^t \sigma_j \cdot \psi_j(z) \cdot \psi_j$.

For $d$-finite Spectrum case, consider the case where $\sigma_j = 0$ for all $j > d$. Then, $\phi(z) = \Pi_d[\phi(z)]$ for any $z \in \mathcal{Z}$. That is, $A_h^\tau$ can be written as

$$A_h^\tau := W_h^\tau - W_h, \quad Z_h = \sum_{\tau \in \mathcal{I}_h} A_h^\tau = \sum_{\tau \in \mathcal{I}_h} W_h^\tau - W_h, \tag{82}$$

where $W_h^\tau = \phi(z_h^\tau)\phi(z_h^\tau)^\top$ as a $d \times d$ matrix, and $W_h = \mathbb{E}_{\bar{\pi}}\left[\phi(z_h)\phi(z_h)^\top\right]$. By the boundness of kernel(i.e. $\sup_{z \in \mathcal{Z}} k(z, z) \leq 1$), we have $\|\phi(z)\|_{\mathcal{H}_k} \leq 1, \forall z \in \mathcal{Z}$. By Jensen's inequality, we have

$$\|\Sigma_h\|_{\mathrm{op}} \leq \mathbb{E}_{\bar{\pi}}\left[\left\|\phi\left(s_h, a_h\right)\phi\left(s_h, a_h\right)^\top\right\|_{\mathrm{op}}\right] \leq 1.$$

Similarly, for all $h \in [H]$ and all $\tau \in \mathcal{I}_h$, as it holds that

$$\|A_h^\tau\|_{\mathrm{op}} \leq \|W_h^\tau\|_{\mathrm{op}} + \|W_h\|_{\mathrm{op}} \leq 2,$$

we have

$$\begin{aligned}
\left\|\mathbb{E}_{\bar{\pi}}\left[Z_h^\top Z_h\right]\right\|_{\mathrm{op}} &= N_H \left\|\mathbb{E}_{\bar{\pi}}\left[\left(A_h^\tau\right)^\top A_h^\tau\right]\right\|_{\mathrm{op}} \\
&\leq N_H \left\|\mathbb{E}_{\bar{\pi}}\left[A_h^\tau \left(A_h^\tau\right)^\top\right]\right\|_{\mathrm{op}} \\
&\leq 4N_H.
\end{aligned}$$

Similarly, we have

$$\left\|\left(A_h^\tau\right)^\top A_h^\tau\right\|_{\mathrm{op}} \leq \left\|\left(A_h^\tau\right)^\top\right\|_{\mathrm{op}} \cdot \|A_h^\tau\|_{\mathrm{op}} \leq 4 \text{ and } \left\|\mathbb{E}_{\bar{\pi}}\left[Z_h^\top Z_h\right]\right\|_{\mathrm{op}} \leq 4N_H.$$

Applying Lemma C.4 for $Z_h$ defined in equation (80), for any fixed $h \in [H]$ and any $l \geq 0$, we have

$$\mathbb{P}\left(\|Z_h\|_{\mathrm{op}} > l\right) = \mathbb{P}\left(\left\|\sum_{\tau \in \mathcal{I}_h} A_h^\tau\right\|_{\mathrm{op}} > l\right) \leq 2d \cdot \exp\left(-\frac{l^2/2}{4N_H + 2l/3}\right).$$

For all $\delta \in (0, 1)$, we set $l = \sqrt{10N_H \log(4dH/\delta)}$, for sufficiently large $N_H \geq 5\log(4dH/\delta)$, we obtain $\|Z_h\|_{\mathrm{op}} \leq \sqrt{10N_H \log(4dH/\delta)}$ holds with probability at least $1 - \delta/2H$.

Moreover, $Z_h$ defined in equation (80) can be written as

$$Z_h = \sum_{\tau \in \mathcal{I}_h} \phi\left(x_h^\tau, a_h^\tau\right) \phi\left(x_h^\tau, a_h^\tau\right)^\top - N_H \cdot \Sigma_h = (\Lambda_h - \lambda \cdot I) - N_H \cdot \Sigma_h. \tag{83}$$

Recall that there exist positive constant $c_{\min}$ such that $\inf_{\|f\|_{\mathcal{H}_k}=1}\langle f, \Sigma_h f\rangle_{\mathcal{H}_k} \geq c_{\min}$. For sufficiently large $N_H \geq \frac{4C^2}{c_{\min}^2}\log\left(4dH/\delta\right)$, we have

$$
\begin{aligned}
\inf_{\|f\|_{\mathcal{H}_k}=1}\langle f, \frac{\Lambda_h}{N_H}f\rangle_{\mathcal{H}_k} &\geq \inf_{\|f\|_{\mathcal{H}_k}=1}\langle f, \left(\frac{Z_h}{N_H} + \Sigma_h + \frac{\lambda}{N_H}I\right)f\rangle_{\mathcal{H}_k} \\
&\geq \frac{1}{N_H}\inf_{\|f\|_{\mathcal{H}_k}=1}\langle f, Z_h f\rangle_{\mathcal{H}_k} + \inf_{\|f\|_{\mathcal{H}_k}=1}\langle f, \Sigma_h f\rangle_{\mathcal{H}_k} \\
&\geq c_{\min} - \frac{1}{N_H}\|Z_h\|_{\mathrm{op}} \geq c_{\min} - C\sqrt{\frac{\log\left(4HG(N_H,\lambda)/\delta\right)}{N_H}} \geq \frac{c_{\min}}{2}.
\end{aligned}
\tag{84}
$$

Hence, it holds that

$$\|\Lambda_h^{-1}\|_{\mathrm{op}} \leq \frac{2}{N_H \cdot c_{\min}}, \tag{85}$$

for all $h \in [H]$. This implies that

$$\|\Lambda_h^{-\frac{1}{2}}\phi(s,a)\|_{\mathcal{H}_k} \leq \|\phi(s,a)\|_{\mathcal{H}_k}\|\Lambda_h^{-1}\|_{\mathrm{op}}^{1/2} \leq c'/\sqrt{N_H}, \tag{86}$$

where $c' = \sqrt{2/c_{\min}}$ and using the fact that $\|\phi(s,a)\|_{\mathcal{H}_k} \leq 1$ for all $(s,a) \in \mathcal{S} \times \mathcal{A}$.

We define the event

$$\mathcal{E}_1^* = \left\{\|\Lambda_h^{-\frac{1}{2}}\phi(s,a)\|_{\mathcal{H}_k} \leq c'/\sqrt{N_H} \text{ for all } (s,a) \in \mathcal{S} \times \mathcal{A} \text{ and all } h \in [H]\right\}.$$

By equation (86), we have $\mathbb{P}(\mathcal{E}_1^*) \geq 1 - \delta/2$ for $N_H \geq \frac{4C^2}{c_{\min}^2}\log\left(4dH/\delta\right)$. Recall that for $d$-finite spectrum case, we have

$$
\begin{aligned}
\beta_h(\delta) &= \sqrt{1 + \frac{1}{N_1}}\mathcal{S} + \sqrt{C_1 \cdot d \cdot \log N_1 + \log(\frac{1}{\delta^2})}, \\
B &= C_2 \cdot H \cdot \sqrt{d\log\left(N/\delta\right)}.
\end{aligned}
\tag{87}
$$

Use big tilde O notation, they can be written as

$$
\begin{aligned}
\beta_h(\delta) &= \tilde{\mathcal{O}}(\sqrt{d}), \\
B &= \tilde{\mathcal{O}}(H\sqrt{d}).
\end{aligned}
\tag{88}
$$

Combining the result in Theorem 4.3 and equation (88) with $\delta = \delta/4$, we have

$$
\begin{aligned}
\mathrm{SubOpt}(\hat{\pi}; s) &\leq 2\sum_{h=1}^{H}\beta_h(\delta)\mathbb{E}_{\pi^*}\left[\|\phi(s_h, a_h)\|_{(\Lambda_h^{\mathcal{D}_1})^{-1}} \mid s_1 = s\right] \\
&\quad + 2B\sum_{h=1}^{H}\mathbb{E}_{\pi^*}\left[\|\phi(s_h, a_h)\|_{(\Lambda_h^{\widetilde{\mathcal{D}}_h^\theta})^{-1}} \mid s_1 = s\right] \\
&\leq 2\beta_h(\delta) \cdot H \cdot c'/\sqrt{N_1} + 2B \cdot H \cdot c'/\sqrt{N_H} \\
&= \tilde{\mathcal{O}}(H\sqrt{\frac{d}{N_1}}) + \tilde{\mathcal{O}}(H^2\sqrt{\frac{d}{N_H}}),
\end{aligned}
\tag{89}
$$

where the last equality follows from the fact that $N_H = N/H$ and $N = N_1 + N_2$ for all $h \in [H]$.

## C   Sufficient Lemma

**Lemma C.1** (Concentration of Self-Normalized Processes in RKHS (Chowdhury & Gopalan, 2017))**.** *Let $\mathcal{H}$ be an RKHS defined over $\mathcal{X} \subseteq \mathbb{R}^d$ with kernel function $k(\cdot, \cdot) : \mathcal{X} \times \mathcal{X} \to \mathbb{R}$. Let $\{x_\tau\}_{\tau=1}^\infty \subset \mathcal{X}$ be a discrete time stochastic process that is adapted to the filtration $\{\mathcal{F}_t\}_{t=0}^\infty$. Let $\{\epsilon_\tau\}_{\tau=1}^\infty$ be a real-valued stochastic process such that (i) $\epsilon_\tau$ is $\mathcal{F}_\tau$-measurable and (ii) $\epsilon_\tau$ is zero-mean and $\sigma$-sub-Gaussian conditioning on $\mathcal{F}_{\tau-1}$, i.e.,*

$$\mathbb{E}\left[\epsilon_\tau \mid \mathcal{F}_{\tau-1}\right] = 0, \quad \mathbb{E}\left[e^{\lambda \epsilon_\tau} \mid \mathcal{F}_{\tau-1}\right] \leq e^{\lambda^2 \sigma^2 / 2}, \quad \forall \lambda \in \mathbb{R}.$$

*Moreover, for any $t \geq 2$, let $E_t = (\epsilon_1, \ldots, \epsilon_{t-1})^\top \in \mathbb{R}^{t-1}$ and $K_t \in \mathbb{R}^{(t-1) \times (t-1)}$ be the Gram matrix of $\{x_\tau\}_{\tau \in [t-1]}$. Then for any $\eta > 0$ and any $\delta \in (0, 1)$, with probability at least $1 - \delta$, it holds simultaneously for all $t \geq 1$ that*

$$E_t^\top \left[(K_t + \eta \cdot I)^{-1} + I\right]^{-1} E_t \leq \sigma^2 \cdot \log \det \left[(1 + \eta) \cdot I + K_t\right] + 2\sigma^2 \cdot \log(1/\delta).$$

*Proof.* Please refer to Theorem 1 in (Chowdhury & Gopalan, 2017) □

**Lemma C.2** (Lemma D.5 in (Yang et al., 2020))**.** *Let $\mathcal{Z}$ be a compact subset of $\mathbb{R}^d$ and $k : \mathcal{Z} \times \mathcal{Z} \to \mathbb{R}$ be the RKHS kernel of $\mathcal{H}$. We assume $k$ is a bounded kernel so that $\sup_{z \in \mathcal{Z}} k(z, z) \leq 1$, and $k$ is continuously differentiable on $\mathcal{Z} \times \mathcal{Z}$. Moreover, let $T_k$ be the integral operator induced by $k$ and the Lebesgue measure on $\mathcal{Z}$, such that $T_k f(z) = \int_{\mathcal{Z}} k(z, z') \cdot f(z') \, dz', \quad \forall f \in \mathcal{L}^2(\mathcal{Z})..$ Let $\{\sigma_j\}_{j \geq 1}$ be the non-increasing sequence of eigenvalues of $T_k$. Recall the definition of maximal information gain in equation (2). We assume $\{\sigma_j\}_{j \geq 1}$ satisfies one of the following eigenvalue decay conditions:*

- *$\gamma$-finite spectrum: $\sigma_j = 0$ for all $j > \gamma$, where $\gamma$ is a positive integer.*

- *$\gamma$-exponential decay: there exists some constants $C_1, C_2 > 0$ such that $\sigma_j \leq C_1 \cdot \exp(-C_2 \cdot j^\gamma)$ for all $j \geq 1$, where $\gamma > 0$ is a positive constant.*

- *$\gamma$-polynomial decay: there exists some constants $C_1 > 0, \tau \in [0, 1/2)$ and $C_\psi > 0$ such that $\sigma_j \leq C_1 \cdot j^{-\gamma}$ and $\sup_{z \in \mathcal{Z}} \sigma_j^\tau \cdot |\psi_j(z)| \leq C_\psi$ for all $j \geq 1$, where $\gamma > 1$.*

*Suppose $\lambda \in [c_1, c_2]$ for absolute constants $c_1, c_2$. Then we have*

$$G(N, \lambda) \leq \begin{cases} C \cdot \gamma \cdot \log N & \gamma\text{-finite spectrum}, \\ C \cdot (\log N)^{1+1/\gamma} & \gamma\text{-exponential decay}, \\ C \cdot N^{(d+1)/(\gamma+d)} \cdot \log N & \gamma\text{-polynomial decay}, \end{cases}$$

*where $C$ is an absolute constant that only depends on $d, \gamma, C_1, C_2, C, c_1$ and $c_2$.*

*Proof.* Please refer to Lemma D.5 in (Yang et al., 2020) for a detailed proof. □

**Lemma C.3** (Extended Value Difference (Cai et al., 2020))**.** *Let $\pi = \{\pi_h\}_{h=1}^H$ and $\pi' = \{\pi'_h\}_{h=1}^H$ be any two policies and let $\left\{\widehat{Q}_h\right\}_{h=1}^H$ be any estimated Q-functions. For all $h \in [H]$, we define the estimated value function $\widehat{V}_h : \mathcal{S} \to \mathbb{R}$ by setting $\widehat{V}_h(x) = \left\langle \widehat{Q}_h(x, \cdot), \pi_h(\cdot \mid x)\right\rangle_{\mathcal{A}}$ for all $x \in \mathcal{S}$. For all $x \in \mathcal{S}$, we have*

$$\widehat{V}_1(x) - V_1^{\pi'}(x) = \sum_{h=1}^H \mathbb{E}_{\pi'} \left[\left\langle \widehat{Q}_h(s_h, \cdot), \pi_h(\cdot \mid s_h) - \pi'_h(\cdot \mid s_h)\right\rangle_{\mathcal{A}} \mid s_1 = x\right]$$
$$+ \sum_{h=1}^H \mathbb{E}_{\pi'} \left[\widehat{Q}_h(s_h, a_h) - \left(\mathbb{B}_h \widehat{V}_{h+1}\right)(s_h, a_h) \mid s_1 = x\right],$$

*where $\mathbb{E}_{\pi'}$ is taken with respect to the trajectory generated by $\pi'$, while $\mathbb{B}_h$ is the Bellman operator defined in Section 3.1.*

*Proof.* Fix $h \in [H]$. Denote that $\iota_i = \widehat{Q}_i - \mathbb{B}_i \widehat{V}_{i+1}$. For all $i \in [h, H]$ and $s \in \mathcal{S}$, we have

$$
\begin{aligned}
&\mathbb{E}_{\pi'} \left[ \widehat{V}_i (s_i) - V_i^{\pi'} (s_i) \mid s_h = s \right] \\
&= \mathbb{E}_{\pi'} \left[ \left\langle \widehat{Q}_i (s_i, \cdot), \pi_i (\cdot \mid s_i) \right\rangle - \left\langle Q_i^{\pi'} (s_i, \cdot), \pi_i' (\cdot \mid s_i) \right\rangle \mid s_h = s \right] \\
&= \mathbb{E}_{\pi'} \left[ \left\langle \widehat{Q}_i (s_i, \cdot), \pi_i (\cdot \mid s_i) - \pi_i' (\cdot \mid s_i) \right\rangle + \left\langle \widehat{Q}_i (s_i, \cdot) - Q_i^{\pi'} (s_i, \cdot), \pi_i' (\cdot \mid s_i) \right\rangle \mid s_h = s \right] \\
&= \mathbb{E}_{\pi'} \left[ \left\langle \widehat{Q}_i (s_i, \cdot), \pi_i (\cdot \mid s_i) - \pi_i' (\cdot \mid s_i) \right\rangle \mid s_h = s \right] \\
&\quad + \mathbb{E}_{\pi'} \left[ \left\langle \iota_i (s_i, \cdot) + \mathbb{B}_i \widehat{V}_{i+1} (s_i, \cdot) - \left( r_i (s_i, \cdot) + \mathbb{P}_i V_{i+1}^{\pi'} (s_i, \cdot) \right), \pi_i' (\cdot \mid s_i) \right\rangle \mid s_h = s \right] \\
&= \mathbb{E}_{\pi'} \left[ \left\langle \widehat{Q}_i (s_i, \cdot), \pi_i (\cdot \mid s_i) - \pi_i' (\cdot \mid s_i) \right\rangle \mid s_h = s \right] + \mathbb{E}_{\pi'} \left[ \iota_i (s_i, a_i) \mid s_h = s \right] \\
&\quad + \mathbb{E}_{\pi'} \left[ \mathbb{P}_i \left( \widehat{V}_{i+1} - V_{i+1}^{\pi'} \right) (s_i, a_i) \mid s_h = s \right] \\
&= \mathbb{E}_{\pi'} \left[ \left\langle \widehat{Q}_i (s_i, \cdot), \pi_i (\cdot \mid s_i) - \pi_i' (\cdot \mid s_i) \right\rangle \mid s_h = s \right] + \mathbb{E}_{\pi'} \left[ \iota_i (s_i, a_i) \mid s_h = s \right] \\
&\quad + \mathbb{E}_{\pi'} \left[ \widehat{V}_{i+1} (s_{i+1}) - V_{i+1}^{\pi'} (s_{i+1}) \mid s_h = s \right],
\end{aligned}
\tag{90}
$$

where $\mathbb{P}_i$ is the transition operator defined in Section 3.1. Rewrite equation (90), we have

$$
\begin{aligned}
&\mathbb{E}_{\pi'} \left[ \widehat{V}_i (s_i) - V_i^{\pi'} (s_i) \mid s_h = s \right] - \mathbb{E}_{\pi'} \left[ \widehat{V}_{i+1} (s_{i+1}) - V_{i+1}^{\pi'} (s_{i+1}) \mid s_h = s \right] \\
&= \mathbb{E}_{\pi'} \left[ \left\langle \widehat{Q}_i (s_i, \cdot), \pi_i (\cdot \mid s_i) - \pi_i' (\cdot \mid s_i) \right\rangle \mid s_h = s \right] + \mathbb{E}_{\pi'} \left[ \iota_i (s_i, a_i) \mid s_h = s \right].
\end{aligned}
\tag{91}
$$

Taking $\sum_{i=h}^{H}$ on equation (91), then

$$
\begin{aligned}
\widehat{V}_h (s) - V_h^{\pi'} (s) &= \sum_{i=h}^{H} \mathbb{E}_{\pi'} \left[ \left\langle \widehat{Q}_i (s_i, \cdot), \pi_i (\cdot \mid s_i) - \pi_i' (\cdot \mid s_i) \right\rangle \mid s_h = s \right] \\
&\quad + \sum_{i=h}^{H} \mathbb{E}_{\pi'} \left[ \widehat{Q}_h (s_i, a_i) - \mathbb{B}_i \widehat{V}_{i+1} (s_i, a_i) \mid s_h = s \right],
\end{aligned}
\tag{92}
$$

letting $h = 1$ completes the proof. $\qquad\square$

**Lemma C.4** (Matrix Bernstein Inequality (Tropp et al., 2015)). *Suppose that $\{A_k\}_{k=1}^{n}$ are independent and centered random matrices in $\mathbb{R}^{d_1 \times d_2}$, that is, $\mathbb{E}[A_k] = 0$ for all $k \in [n]$. Also, suppose that such random matrices are uniformly upper bounded in the matrix operator norm, that is, $\|A_k\|_{op} \leq L$ for all $k \in [n]$. Let $Z = \sum_{k=1}^{n} A_k$ and*

$$
v(Z) = \max \left\{ \left\| \mathbb{E} \left[ ZZ^\top \right] \right\|_{op}, \left\| \mathbb{E} \left[ Z^\top Z \right] \right\|_{op} \right\} = \max \left\{ \left\| \sum_{k=1}^{n} \mathbb{E} \left[ A_k A_k^\top \right] \right\|_{op}, \left\| \sum_{k=1}^{n} \mathbb{E} \left[ A_k^\top A_k \right] \right\|_{op} \right\}.
$$

*For all $t \geq 0$, we have*

$$
\mathbb{P} \left( \|Z\|_{op} \geq t \right) \leq (d_1 + d_2) \cdot \exp \left( -\frac{t^2/2}{v(Z) + L/3 \cdot t} \right).
$$

*Proof.* See Tropp et al. (2015, Theorem 1.6.2) for a detailed proof. $\qquad\square$

