# OpenReview forum: "Leveraging Unlabeled Data Sharing through Kernel Function Approximation in Offline Reinforcement Learning"
_TMLR — Accepted by TMLR_

### Review · Reviewer_CUaV · 2024-08-27

**Summary Of Contributions:**

The paper considers enhancing offline reinforcement learning (RL) by incorporating unlabeled (no rewards) offline data. The authors propose an algorithm that utilizes kernel function approximation to leverage unlabeled data, particularly when obtaining labeled data is costly or limited. The paper provides theoretical guarantees for the algorithm's performance and discusses various conditions of eigenvalue decay and coverage of the offline dataset that determine the algorithm's overall complexity.

**Audience:**

Yes

**Claims And Evidence:**

Yes

**Requested Changes:**

See above.

**Strengths And Weaknesses:**

Strengths:

- The paper addresses a significant problem in offline reinforcement learning by exploring how unlabelled data can be leveraged to enhance performance.

- It extends the PDS framework to kernel settings, broadening the applicability of the PDS framework.

- The paper provides an insightful theoretical analysis, and demonstrating that the assumption of weak convergence in offline RL may conflict with the kernel regularity assumption typically used in online RL.

Weaknesses:

- The paper lacks empirical validation. While it is primarily theoretical, including experiments on illustrative problems would help demonstrate the effectiveness of the proposed kernel-based method. Additionally, a more detailed discussion on the implementation aspects, such as how to compute the pessimistic reward function in kernel settings, would be beneficial.

- The related work section is somewhat limited in scope. For instance, there are several works on offline RL theory that explore general function approximation, which are not adequately discussed, such as [1,2,3,4].

- The paper would benefit from a more thorough discussion of the assumptions. For example, providing concrete examples where the conditions outlined in Assumption 3.1 hold would enhance the clarity and applicability of the theoretical results.

- The discussion following Corollary 4.9 regarding the assumptions on weak convergence and kernel regularity is intriguing but lacks depth. Expanding this discussion, particularly on the type of assumptions required for the offline data coverage to achieve concrete performance bounds in cases of $d$-exponential decay and $d$-polynomial decay, would significantly strengthen the paper.


[1] Uehara, Masatoshi, and Wen Sun. "Pessimistic model-based offline reinforcement learning under partial coverage." arXiv preprint arXiv:2107.06226 (2021).

[2] Yin, Ming, Mengdi Wang, and Yu-Xiang Wang. "Offline reinforcement learning with differentiable function approximation is provably efficient." arXiv preprint arXiv:2210.00750 (2022).

[3] Blanchet, Jose, et al. "Double pessimism is provably efficient for distributionally robust offline reinforcement learning: Generic algorithm and robust partial coverage." Advances in Neural Information Processing Systems 36 (2024).

[4]  Hu, Hao, et al. "Bayesian Design Principles for Offline-to-Online Reinforcement Learning." arXiv preprint arXiv:2405.20984 (2024).

---

> ### Author Response · Authors · 2025-02-02
> **Response to Reviewer CUaV**
>
> We appreciate your thoughtful feedback, which helps refine our work. In response, we include empirical validation, clarify implementation details, and expand discussions on assumptions and related work.
>
> **Weaknesses**
>
> *The paper lacks empirical validation. While it is primarily theoretical, including experiments on illustrative problems would help demonstrate the effectiveness of the proposed kernel-based method. Additionally, a more detailed discussion on the implementation aspects, such as how to compute the pessimistic reward function in kernel settings, would be beneficial.*
>
> **Response**: Thank you for your suggestions. In our updated version, we provides two experiments, the first experiment show the actual asymptotic behavior of $V^{\\pi}(s)$ will be proportional to $N\_1^{-1/2}$ and $N\_2^{-1/2}$, as predicted by our theorem. The second one shows our method compared with finite-dimensional features, which was proposed by the previous work. Our method shows superiority compared to finite-dimensional features. Besides, we also introduce the implementation aspect including pessimistic reward, as well as provide the source code in supplementary materials.
>
> *The related work section is somewhat limited in scope. For instance, there are several works on offline RL theory that explore general function approximation, which are not adequately discussed, such as \[1,2,3,4\].*
>
> **Response**: Thank you for your valuable comment. In the updated version of our paper, we added these paper to the discussion of related works.
>  - For \[1\], they focus on model-based reinforcement learning, which differs from our assumption of a model-free RL setting.
>  - For \[2\], they study offline reinforcement learning with differentiable function class approximations, whereas we consider general function classes.
>  - For \[3\], they focus on robust MDPs, while we consider non-robust MDPs, specifically aiming to find an optimal policy given an offline dataset collected a priori.
>  - For \[4\], they assume the MDP can be described by an unknown model parameter in the linear MDP setting to bridge offline-to-online reinforcement learning. However, we focus exclusively on offline RL.
>
> *The paper would benefit from a more thorough discussion of the assumptions. For example, providing concrete examples where the conditions outlined in Assumption 3.1 hold would enhance the clarity and applicability of the theoretical results.*
>
> **Response**:  Thank you for your valuable comment. To enhance the clarity and applicability of Assumption 3.1, we provide the following examples:
> - $\\gamma$-exponential decay: This condition is satisfied by the squared exponential kernel.
> - $\\gamma$-polynominal decay: This condition holds for the Matérn kernel with $\\nu \> 1$.
>
> The detailed formulations for these cases are included in the paper for reference.  [https://arxiv.org/pdf/0912.3995](https://arxiv.org/pdf/0912.3995)
>
> *The discussion following Corollary 4.9 regarding the assumptions on weak convergence and kernel regularity is intriguing but lacks depth. Expanding this discussion, particularly on the type of assumptions required for the offline data coverage to achieve concrete performance bounds in cases of $d$-exponential decay and $d$-polynomial decay, would significantly strengthen the paper.*
>
> **Response**: We appreciate the reviewer’s comment and have provided an explanation in Remark 4.9 regarding why $d$\-exponential decay and $d$\-polynomial decay are not discussed. Specifically, these decay assumptions are highly restrictive and may not align with the generality of our framework.

---

### Review · Reviewer_HXF3 · 2024-10-31

**Summary Of Contributions:**

This paper explores the challenges of offline reinforcement learning (RL), which relies on a static dataset for policy learning but typically demands extensive data. The paper addresses the issue of high costs associated with obtaining large labeled datasets, particularly when human labelers are required to provide rewards. The authors emphasize the value of leveraging unlabeled data, which is more cost-effective, in offline RL scenarios where labeled data is either scarce or prohibitively expensive.

The paper introduces an algorithm that incorporates unlabeled data into offline RL using kernel function approximation and offers theoretical guarantees for its effectiveness. It discusses various conditions related to eigenvalue decay of the Hk matrix, which influence the algorithm's complexity. Overall, the study presents a promising method for capitalizing on the benefits of unlabeled data in offline RL while ensuring theoretical assurances of performance.

**Audience:**

Yes

**Claims And Evidence:**

Yes

**Requested Changes:**

1. If possible, a few small experiments could be added to validate the theoretical results.

2. I have no additional suggestions for revising this paper. Considering the importance of the research direction, the solid theoretical analysis, and the interesting theoretical results of this work, I recommend accepting the paper.

**Strengths And Weaknesses:**

Strengths:

1. This paper is well-written and easy to follow.

2. The theoretical analysis in this paper is very solid.

3. This paper investigates an important issue, the data utilization problem in offline RL. This paper extends an important method, PDS, to a broader range of scenarios.

Weaknesses:

1. This paper is purely theoretical analysis and lacks some experiments for validation.

---

> ### Author Response · Authors · 2025-02-02
> **Response to Reviewer HXF3**
>
> We sincerely appreciate your thoughtful feedback, which has been instrumental in refining our work. In response, we have included empirical validation.
>
> In our revised version, we present two experiments. The first demonstrates that the asymptotic behavior of $V^{\\pi}(s)$ is proportional to $N_1^{-1/2}$ and $N_2^{-1/2}$, aligning with our theoretical predictions. The second experiment compares our method with finite-dimensional features introduced in previous work, showing that our approach outperforms these alternatives.
> Additionally, we discuss implementation aspects, including pessimistic reward estimation, and provide the source code in the supplementary materials.

---

> > ### Comment · Reviewer_HXF3 · 2025-02-05
> > **response**
> >
> > Thanks for your replay. My concerns have been solved.

---

### Review · Reviewer_Qc77 · 2025-01-05

**Summary Of Contributions:**

The paper studies how to leverage unlabeled data (i.e., only has transition information but does not have reward information) in the offline RL setting, which is practical given more and more video demonstrations and human labeling is expensive. The hope is that, with help from a small amount of labeled data, we can label a large amount of unlabeled data to reduce the amount of labeled data. The previous paper considers linear mdp setting, and they proposed the unsupervised data sharing method where one first fits a reward function on the labeled data and then use it to label the unlabeled data. Compared with the canonical baseline, the requirement on the labeled data is reduced by $O(H \sqrt{d})$.

This paper extends the unsupervised data sharing method into the kernel setting and also incorporates it with the data splitting scheme. It archives guarantees both on the d-finite spectrum kernel setting and general kernel setting, and when reduced to linear mdp setting, it compares with the previous method with a $H$ and $d$ tradeoff.

**Audience:**

Yes

**Claims And Evidence:**

Yes

**Requested Changes:**

1. I think the paper needs to highlight the real contribution of the paper, i.e., what is the new assumption considered, and how do they really compares with the previous literature, and more importantly, what are the new techniques introduced in the paper what would possible benefit the community. From the current writing this is unclear to me.
2. As the paper only considers the kernel function approximation setting, I think it would be beneficial to move the structure assumption (beginning of section 4.1) and the combined structure and function approximation assumption (assumption 4.3) before the introduction of the analysis and even the algorithm, otherwise it confuses the reading while reading the algorithmic choices.
3. (I don't think this is necessary but) the authors should put an effort into at least removing the suboptimal $\sqrt{H}$ introduced by the naive application of the data splitting trick.
4. nit: $\mathcal{H}$ is not defined in the abstract; there are some inconsistencies across the paper on different notations of $\theta$, for example, at the end of section 3.2, in the definition of $D^\theta_2$, should it be $D^{\hat{\theta}}_2$?

**Strengths And Weaknesses:**

## Strength
1. On the technical part, the paper clearly discussed the connection to previous work, for example, the usage of each technique is introduced in which previous paper, the assumption of the kernel function approximation and the eigenvalue decay assumption and their connection to the previous papers, so that the reader can better contextualize the results.
2. The paper is technically sound, with the generalization from linear mdp to kernel function approximation.
3. The paper proposes a sharper concentration of kernel ridge regression (proposition 4.1) for the estimation of the learned reward function, which shares the similar spirit of the previous sharp concentration on the linear ridge regression studied for linear-ucb. But I am a little bit skeptical that this is never proved in the kernel bandit setting.

## Weakness
Overall I am a little bit doubtful about the list of contributions that the authors claimed:
1. About the second contribution, focus on finite horizon setting: maybe I am missing something, but what is the technical significance of the finite vs. discounted setting in the proposed algorithm? Or on the other hand, does the previous Hu et al. paper not trivially extend to the finite horizon setting?
2. The third contribution, the weaker coverage assumption, if I understand correctly, the coverage assumption required by Hu et al. seems only concerns that the offline data cover the optimal policy's traces, instead of a global coverage assumption. However, the proposed coverage assumption 4.8 seems to require certain global coverage assumption in the sense that it needs to cover all direction in the RKHS that we care about?
3. I am also confused about listing the data splitting trick as a contribution, as mentioned in the paper, it is used by previous Xie et al. paper to obtain a sharper rate in offline RL. However, the previous analysis is more careful so that it avoids the $d$ vs. $H$ tradeoff (although it is in the tabular setting but there was no tradeoff), but the current paper seems to just apply the trick naively and maybe caused the suboptimal rate in $H$. I think if this is listed as a contribution, then the analysis needs to be conducted more carefully.
4. One thing that is not super clear to me is that what is the point of relabeling the labeled offline data after the learned reward function. My guess is that if you do not relabel $D_1$, there is still no difference in the rate? But it would be nice to have a short discussion on this matter.
5. Finally this is a critique on the unsupervised data sharing method overall: I think the point of the setting is such that we can have a better rate on the labeled offline data. But as far as I know, both the rate of the previous linear mdp paper, or the rate of this paper reduced to the linear mdp setting, are no better than the currently sharpest rate on the canonical offline rl with linear mdp (for example, [1]). Thus unless a sharper analysis is conducted (which I think is very likely), I do not see much contribution of the current works.

### Reference
[1] Xiong, Wei, et al. "Nearly minimax optimal offline reinforcement learning with linear function approximation: Single-agent mdp and markov game." arXiv preprint arXiv:2205.15512 (2022).

---

> ### Author Response · Authors · 2025-02-02
> **Response to Reviewer Qc77**
>
> Thank you for your detailed and constructive feedback on our paper. Your insights help us refine our explanations and strengthen the presentation of our findings. Below, we address each of your concerns, providing additional explanations, comparisons with prior literature, and clarifications regarding our contributions.
>
> **Weakness**
>
> 1. To ensure convergence under the discounted setting, a stronger structural assumption is necessary. In our case, both the transition function and reward function depend on the horizon, which introduces additional complexity. Therefore, under the discounted setting, we must impose stricter assumptions to address this dependency. In contrast, the paper by Hu et al. assumes that the transition and reward functions are horizon-independent, which simplifies their analysis. This fundamental difference means their approach cannot be directly extended to accommodate our setting.
> 2.  Our assumptions ensure simultaneous comparisons across all policies, offering a stronger guarantee than merely competing with the optimal policy, as proposed by Hu et al.
> 3. Thank you for raising this concern. We acknowledge that the data-splitting trick was introduced in prior work by Xie et al. However, our application of the data-splitting method introduces a trade-off characterized by $\\sqrt{\\frac{H}{d}}$, where the horizon $H$ is fixed, and the value of $d$ is determined by the kernel choice. In the linear setting, when transforming the feature mapping from a dimensionality of $d$ to $d^{\\prime}$ (with $d^\\prime \> d$), the data-splitting trick becomes particularly beneficial. It mitigates the impact of increased dimensionality on the convergence of the error bound, enhancing the learning performance in kernel-based settings.
> 4.  As noted in Hu et al., we retain the pessimistic property of the offline algorithm by relabeling the data. However, since the true reward function is unknown, it becomes challenging to establish a precise bound for our algorithm in the analysis. This limitation arises from the uncertainty in accurately characterizing the reward function's contribution to the overall performance.
> 5. In prior work, such as \[1\], the focus was on achieving better rates by ensuring the dataset size exceeds a sufficient threshold. In contrast, our setting emphasizes the distinction between labeled and unlabeled datasets. In real-world scenarios, obtaining a labeled dataset is costly and challenging, whereas unlabeled datasets are more readily available. Our algorithm addresses this limitation by demonstrating that, with a limited labeled dataset, leveraging the unlabeled dataset can achieve performance comparable to their results.
>
> ### **Requested Changes:**
>
> 1.  Thank you for your insightful comment. To address this, we have revised and clarified the contributions of our paper as follows:
>       1. Extension of PDS framework
>       2. Focus on finite-horizon MDPs
>       3. Feature coverage assessment via concentratability coefficie
>       4. Enhance the suboptimality
>       Further details and comparisons with previous literature are provided at the end of Chapter 1\.
> 2.  Thank you for your insightful comment. We have revised the structure of the paper by rearranging the order of the assumptions. Specifically, we moved the structure assumption and the combined structure and function approximation assumption to appear earlier in the paper, prior to the introduction of the analysis and the algorithm. This reorganization improves the logical flow and enhances the clarity of the algorithmic choices.
> 3. In our setting, the data-splitting method introduces a trade-off between the horizon $H$ and the proxy dimension $d$ of the RKHS $\\mathcal{H}\_k$. By selecting an appropriate kernel, we can sufficiently enhance the suboptimality bound, improving it by a factor of $\\sqrt{d}$. This careful choice of kernel helps balance the trade-off while maintaining performance guarantees.
> 4. Thank you for the clarification. In this context,  $\\mathcal{D}^\\theta\_2$  represents the dataset $\\mathcal{D}$ that has been relabeled using the parameter $\\theta$.

---

> > ### Comment · Reviewer_Qc77 · 2025-03-07
> >
> > 1.  It would be great if the paper can spell out the extra structural assumption required in the discounted setting which could be a motivation for the non-stationary setting studied in this work. It will also be great for the paper to further highlight the additional complexity introduced by the finite horizon setting. As far as I know, in general the analysis for discounted case is usually no easier than the finite horizon setting (e.g. the FQI analysis).
> >
> > 2. The issue of all policy coverage is not that it offers stronger guarantee, but it is a stronger assumption than the single policy coverage.
> >
> > 3. I agree with the author's motivation, but it would be a good technical contribution to avoid the tradeoff, which seems possible given the success in the non-kermel setting.
> >
> > 4. Thank you for the explanation, this makes sense.
> >
> > 5. Thank you for the explanation. My concern is that even the number of labeled data in the setting this paper considers seems no better than just using the sharpest analysis in the regular offline RL setting. I think the analysis should be more meaningful if the result shows that the number of labeled data required is strictly smaller than the number of total data required in the regular offline setting.

---

### Decision · Action_Editor_vuUn · 2025-03-18

**Recommendation:** Accept with minor revision

**Comment:**

Among the detailed reviews, one was more critical of this paper, while one was somewhat positive. On the positive side, as mentioned by Reviewer CUaV, "extends the PDS framework to kernel settings, broadening the applicability of the PDS framework." As mentioned under "Audience", the use of unlabeled data to get better theoretical performance is of sufficient interest to the RL Theory community.

On the negative side, some of the author responses to Reviewer Qc77’s questions did not get at the heart of the questions, which makes it difficult give fairly assess this work in its current form.

First, the reviewer asked why the previous work of Hu et al. does not extend to the finite-horizon setting. The authors respond by first saying that the discounted setting requires a stronger structural assumption (what is this structural assumption?), and then they say that their (the authors') use of non-stationary transition and reward functions makes things more complicated. I cannot draw any clear conclusion from this response. Are the authors saying that the discounted setting is more difficult, or that their (non-stationary) setting is more difficult? In your revision, please carefully (and by all means, clearly) address this comment by Reviewer Qc77:
>> It would be great if the paper can spell out the extra structural assumption required in the discounted setting which could be a motivation for the non-stationary setting studied in this work. It will also be great for the paper to further highlight the additional complexity introduced by the finite horizon setting. As far as I know, in general the analysis for discounted case is usually no easier than the finite horizon setting (e.g. the FQI analysis).

Second, Reviewer Qc77 asked the authors about their Assumption 4.7 (originally Assumption 4.8), wondering if this is a global coverage assumption (which it seems would be a stronger assumption than single policy coverage). The authors response did not address this question. Instead, the authors responded by saying that their assumption allows to compare across all policies, which I stress again is not a proper response to the question. In your revision, please clearly discuss whether your Assumption 4.7 is a global coverage assumption.

Also, from my read of the paper, your theoretical guarantee (Theorem 4.3) only compares against the optimal policy. I apologize if I missed something. If you wish to claim as an advantage that you compare to all policies, then I think you should state the result in a more general way (which, in particular, shows better guarantees for comparing against non-optimal policies, presumably with a bound that depends on some characteristic of the policy being compared to).

I think the other questions from the reviewer were adequately addressed, and I tend to agree with the authors regarding the distinction between labeled and unlabeled datasets.

I am willing to accept this paper with a "minor revision", but I stress that "minor" is nominal. I want to see the authors very well address the points mentioned above, and this will be a condition for acceptance. Revising in this way should improve how your work is received and will ultimately benefit you.

**Audience:**

The use of unlabeled data to get better theoretical performance is of sufficient interest to the RL Theory community. However, in its present form, the paper may have too limited of an audience. I think this can be fixed via a semi-major (and yet, not "major") revision. For details, see "Comment" below.

**Claims And Evidence:**

The theoretical results are technically sound, and the authors have provided sufficient experiments to address one reviewer's concerns.

---

> ### Author Response · Authors · 2025-03-30
> **Response to Action Editor vuUn (Part 1)**
>
> *First, the reviewer asked why the previous work of Hu et al. does not extend to the finite-horizon setting. The authors respond by first saying that the discounted setting requires a stronger structural assumption (what is this structural assumption?), and then they say that their (the authors') use of non-stationary transition and reward functions makes things more complicated. I cannot draw any clear conclusion from this response. Are the authors saying that the discounted setting is more difficult, or that their (non-stationary) setting is more difficult?*
> - Thank you for the helpful comments. Below is a more detailed explanation, which distinguishes between the structural assumptions and the non-stationary aspects in our work, and clarifies which of these elements presents greater difficulty.
> 	- Structural Assumptions
> 		- Hu et al.’s finite coverage coefficient:  Hu et al. rely on a finite coverage coefficient (a concentrability condition), which ensures sufficient exploration and sample coverage in the discounted setting. This is a structural assumption on the data distribution.
> 		- Assumptions in our main theorem (Theorem 4.3): Our main theorem relies on Assumptions 3.1 and 3.2, which are mild and standard. Assumption 3.1 is a classic condition in RKHS analysis, and Assumption 3.2 holds trivially for linear MDPs [2, 3]. These assumptions are structural, as they set the conditions for the RKHS-based analysis.
> 		- Additional Assumption 4.7 for RKHS: For the infinite-dimensional RKHS case, we introduce Assumption 4.7, which is stronger than the finite coverage coefficient in Hu et al.’s work. This assumption is necessary because standard concentrability conditions do not extend well to infinite-dimensional spaces. This is also a structural assumption.
> 	- Non-Stationary Aspects
> 		- Finite-horizon setting without uniform coverage: Our finite-horizon setting does not require the uniform coverage assumption that Hu et al. use. This reflects a shift from a structural assumption to a different approach that accommodates non-stationary environments.
> 		- Time-dependent transitions and rewards: A key feature of our finite-horizon setting is the presence of time-dependent transitions and rewards, which introduces non-stationary dynamics. This aspect of the problem significantly complicates the theoretical analysis because error propagation is more challenging without contraction properties.
> 	- Which is More Difficult?
> 		- Stationary and Finite-Dimensional RKHS: In the case of stationary and finite-dimensional RKHS, the finite-horizon setting and the discounted setting are quite similar in terms of complexity. There is no significant difference between the two in this context.
> 		- Infinite-Dimensional RKHS: In the infinite-dimensional RKHS setting, the discounted setting requires additional assumptions, as highlighted in [1], particularly to account for contraction properties and the challenges posed by infinite-dimensional function spaces. On the other hand, the finite-horizon setting does not rely on these additional assumptions and can be more straightforward to analyze.
> 		- Non-Stationary Finite-Dimensional and Infinite-Dimensional RKHS: For non-stationary settings in either finite or infinite-dimensional RKHS, the finite-horizon setting is more tractable than the discounted setting. In our work, we fully address the finite-horizon case in finite-dimensional RKHS. The infinite-dimensional case, especially for exponential and polynomial decay, is discussed in Remark 4.9. For the non-stationary discounted setting, more structured assumptions are needed, making it more challenging to analyze
>
> **Reference**
> 1. Nguyen-Tang, Thanh, et al. "Sample complexity of offline reinforcement learning with deep ReLU networks." arXiv preprint arXiv:2103.06671 (2021).
> 2. Qi Cai, Zhuoran Yang, Chi Jin, and Zhaoran Wang. Provably efficient exploration in policy optimization. In International Conference on Machine Learning, pp. 1283–1294. PMLR, 2020.
> 3. Andrea Zanette, Ching-An Cheng, and Alekh Agarwal. Cautiously optimistic policy optimization and exploration with linear function approximation. In Conference on Learning Theory, pp. 4473–4525. PMLR, 2021.
> 4. Ming Yin, Yaqi Duan, Mengdi Wang, and Yu-Xiang Wang. Near-optimal offline reinforcement learning with linear representation: Leveraging variance information with pessimism. In International Conference on Learning Representations, 2022a.
> 5. Andrew Wagenmaker and Aldo Pacchiano. Leveraging offline data in online reinforcement learning. In International Conference on Machine Learning, pp. 35300–35338. PMLR, 2023.

---

> ### Author Response · Authors · 2025-03-30
> **Response to Action Editor vuUn (Part 2)**
>
> *Second, Reviewer Qc77 asked the authors about their Assumption 4.7 (originally Assumption 4.8), wondering if this is a global coverage assumption (which it seems would be a stronger assumption than single policy coverage). The authors response did not address this question. Instead, the authors responded by saying that their assumption allows to compare across all policies, which I stress again is not a proper response to the question. In your revision, please clearly discuss whether your Assumption 4.7 is a global coverage assumption.*
> -  We appreciate the reviewer and editor for raising this important clarification. Assumption 4.7 indeed corresponds to a global coverage assumption over the entire policy class, rather than a per-policy coverage assumption.Specifically, this means the dataset must ensure sufficient coverage over all state-action pairs that could be visited by any policy within the considered class. This requirement is more stringent compared to single-policy coverage, as it demands statistical guarantees for the entire policy space.
> - To further analyze the role of data coverage,  we introduce Corollary 4.8, which explores the convergence behavior of Algorithm 1 under a relaxed data coverage assumption. During our research, we initially attempted to apply the coverage assumption from Hu et al., but encountered significant challenges in extending it to the infinite-dimensional RKHS.
> - For the finite-spectrum case (linear MDP setting), we can substantially relax this global coverage assumption. However, when kernel eigenvalues demonstrate exponential or polynomial decay, more robust assumptions become necessary to establish theoretical convergence guarantees, as elaborated in Remark 4.9. We will enhance the paper to explicitly delineate these theoretical nuances.
> - In our future work, we aim to explore the application of this assumption in infinite-dimensional RKHS settings.
>
> **Reference**
> 1. Nguyen-Tang, Thanh, et al. "Sample complexity of offline reinforcement learning with deep ReLU networks." arXiv preprint arXiv:2103.06671 (2021).
> 2. Qi Cai, Zhuoran Yang, Chi Jin, and Zhaoran Wang. Provably efficient exploration in policy optimization. In International Conference on Machine Learning, pp. 1283–1294. PMLR, 2020.
> 3. Andrea Zanette, Ching-An Cheng, and Alekh Agarwal. Cautiously optimistic policy optimization and exploration with linear function approximation. In Conference on Learning Theory, pp. 4473–4525. PMLR, 2021.
> 4. Ming Yin, Yaqi Duan, Mengdi Wang, and Yu-Xiang Wang. Near-optimal offline reinforcement learning with linear representation: Leveraging variance information with pessimism. In International Conference on Learning Representations, 2022a.
> 5. Andrew Wagenmaker and Aldo Pacchiano. Leveraging offline data in online reinforcement learning. In International Conference on Machine Learning, pp. 35300–35338. PMLR, 2023.